# Causes of the 2015 North Atlantic cold anomaly in the ECCOv4 state estimate

Rachael N.C. Sanders[1], Daniel C. Jones[1], Simon A. Josey[2], Bablu Sinha[2], and Gael Forget[3]

[1]British Antarctic Survey, NERC, UKRI, Cambridge, UK
[2]National Oceanography Centre, Southampton, UK
[3]EAPS, MIT, Cambridge, MA, USA

**Correspondence:** Rachael Sanders (racnde@bas.ac.uk)

**Abstract.** The subpolar North Atlantic is an important part of the global ocean and climate system, with SST variability in the region influencing the climate of Europe and North America. While the majority of the global ocean exhibited higher than average surface temperatures in 2015, the subpolar North Atlantic experienced record low temperatures. This interannual cold anomaly is thought to have been driven by surface forcing, but detailed questions remain about how the anomaly was created and maintained. To better quantify and understand the processes responsible for the cold anomaly, we computed mixed layer temperature budgets in the ECCO Version 4 global ocean state estimate. State estimates have been brought into consistency with a large suite of observations without using artificial sources or sinks of heat, making them ideal for temperature budget studies. We found that strong surface forcing drove approximately 75% of the initial anomalies in the cooling of the mixed layer in December 2013, while horizontal advection drove the remaining 25%. The cold anomaly was then sequestered beneath the mixed layer. Re-emergence of the cold anomaly during the summer/autumn of 2014 was primarily the result of a strong temperature gradient across the base of the mixed layer, with vertical diffusion accounting for approximately 70% of the re-emergence. Weaker surface warming of the mixed layer during the summer of 2015 enhanced the anomaly, causing a temperature minimum. Spatial patterns in the budgets also show large differences between the north and south of the anomaly region, with particularly strong initial surface cooling in the south related to the positive phase of the East Atlantic Pattern. It is important to note that this interannual cold anomaly, which is thought to be primarily driven by surface forcing, is distinct from the multi-decadal North Atlantic "warming hole", which has been associated with changes in advection.

## 1 Introduction

In 2015, while the majority of the global ocean experienced warmer than average surface temperatures, the subpolar North Atlantic instead experienced record low temperatures. This phenomenon, often described as the "Atlantic cold blob" is thought to have been driven primarily by surface forcing. The coldest monthly anomalies occurred during the summer of 2015, with sea surface temperatures (SSTs) reaching around 2°C lower than the long-term average (Duchez et al., 2016; de Jong and de Steur, 2016). The anomaly extended across the subpolar North Atlantic and was observed from the surface to depths of at least 500 m (Duchez et al., 2016; Josey et al., 2018). The low temperatures led to an increase in convection (Piron et al., 2017) and enhanced formation of Subpolar Mode Water (Grist et al., 2016), with subsequent effects on the local climate (Duchez

et al., 2016; Mecking et al., 2019) and surrounding ecosystems (Hátún et al., 2017). The rarity and intensity of the 2015 cold
anomaly event, as well as the rate at which the anomaly formed, mean that such events are difficult to predict using models
(Maroon et al., 2021). The Atlantic cold blob is separate to a multi-decadal cooling trend that has also been observed in the
North Atlantic (Drijfhout et al., 2012; Rahmstorf et al., 2015).

Previous studies have highlighted the importance of anomalous surface heat loss in driving anomalies in both surface tem-
peratures and the strength of the convection in the North Atlantic (de Jong and de Steur, 2016; Yeager et al., 2016; Desbruyères
et al., 2019; Kostov et al., 2021). From 2014-2015, the subpolar North Atlantic experienced the strongest surface heat loss
since the 1980s (Yeager et al., 2016). This extreme heat loss spanned the whole subpolar gyre (Piron et al., 2017) and has been
linked to anomalously strong westerly and northerly winds transporting colder air over the region (Grist et al., 2016). While
there is a strong relationship between variability in upper ocean temperature and the strength of the Atlantic overturning circu-
lation (Desbruyères et al., 2019; Kostov et al., 2021), surface forcing from 2013-14 was strong enough to erode any correlation
between the strength of the overturning and the upper ocean heat content (Desbruyères et al., 2019).

Re-emergence, the process by which surface temperature anomalies are "stored" beneath the mixed layer and later brought
back up to the surface again as the mixed layer deepens, has also been shown to be important for driving and sustaining
temperature anomalies over consecutive years (Alexander et al., 1999). In the North Atlantic, this process involves surface-
driven SST anomalies associated with atmospheric modes of variability, such as the North Atlantic Oscillation (NAO) or East
Atlantic Pattern (EAP), being sequestered beneath the seasonal thermocline as the mixed layer shallows during spring/summer.
The anomalies then re-emerge at the surface the following autumn/winter, as the mixed layer deepens again (Cassou et al.,
2007; Taws et al., 2011).

The 2015 cold anomaly has been linked to the two leading North Atlantic atmospheric modes of variability. The NAO is
defined by the pressure gradient between the Iceland Low and Azores High, with a positive index representing a stronger
gradient (Rogers, 1984; Lamb and Peppler, 1987). The EAP is recognised by a pressure anomaly in the east of the subpolar
gyre, with a negative anomaly associated with a positive EAP index (Wallace and Gutzler, 1981; Barnston and Livezey, 1987).
During the strong surface heat loss in the winter of 2013/2014, the EAP was dominant and in its positive phase, while the NAO
was dominant and also positive during the winter of 2014/15 (Yeager et al., 2016; Josey et al., 2018). Conversely, variability
in the surface temperature of the North Atlantic also has a strong influence on the regional climate, as anomalous SSTs
drive changes in atmospheric temperature and subsequent changes in the atmospheric flow (Sutton and Mathieu, 2002). Cold
anomalies in the North Atlantic subpolar gyre have also been linked to European heatwaves, and the 2015 cold anomaly may
have contributed to the development of extreme heatwave conditions in central Europe during the summer of 2015 (Duchez
et al., 2016; Mecking et al., 2019).

Grist et al. (2016) showed that anomalous surface cooling was important in driving the initial cold anomaly from 2013/14,
while Josey et al. (2018) reviewed the various processes involved in driving and sustaining the cold anomaly from 2013 to
2016. In this work, we use mixed layer temperature budgets within an ocean state estimate to quantify the proportion of initial
cooling due to each process, and determine the individual processes driving the re-emergence of the cold anomaly the following

year. We also look at the spatial patterns in the drivers of the cold anomaly, focusing particularly on the differences between the north and south of the cold anomaly region.

Following the methods of previous studies (Frankignoul, 1985; Peter et al., 2006; Dong et al., 2007), we choose to approximate the mixed layer temperature budgets offline using the monthly averaged model output, assigning changes in temperature to well known ocean processes. Because our chosen method uses a certain set of well-understood assumptions, including the concepts of entrainment and lateral induction, it provides unique insights into the evolution of the mixed layer that would be unclear or unavailable in a closed-budget representation. This may sound counter-intuitive, since closed budgets are desirable in a large number of applications. However, the advantage comes from the fact that entrainment and lateral induction represent the average effect of how the temporally-varying mixed layer interacts with its environment over a chosen time period, in our case one month, in a way that is not captured by following the mixed layer at each timestep. However, the disadvantage of this method is that the budgets do not close perfectly. This is due in part to the computation of the budgets on lower temporal resolution data than when closed budgets are computed online. Error is also introduced by the various assumptions that must be made, such as the values chosen to represent diffusivity, and the definition of entrainment velocity. The view of the mixed layer produced by this method should be considered as one among many, as different views will complement each other and help us build a more complete understanding of mixed layer evolution. That being said, for validation purposes, we did compare the similarities between the anomalies in our mixed layer budgets and those in the fully closed budgets, and we found them to be similar.

In Section 2, we discuss the state estimate used to analyse the 2015 cold anomaly, and outline the method used to compute the mixed layer temperature budgets within the model. In Section 3.1, the cold anomaly is analysed within observations in order to validate the cold anomaly within the model in Section 3.2. In Section 3.3, we determine the dominant processes driving seasonal temperature variability within the cold anomaly region, before examining anomalies in these processes in the lead up to, and during, the 2015 cold anomaly in Section 3.4. Finally, in Sections 3.5 and 3.6, the spatial patterns in the processes driving the cold anomaly are explored, focusing particularly on the differences between the north and south of the region.

## 2 Methods and data

### 2.1 Ocean state estimates

We use an MITgcm-based global ocean state estimate to investigate the drivers of the 2015 North Atlantic cold anomaly: Estimating the Circulation and Climate of the Ocean (ECCO) Version 4, Release 4 (ECCOv4-r4, covering 1992-2017) (Fukumori et al., 2017). A *state estimate* is a numerical simulation of the time-evolving ocean state that has been brought into consistency with a suite of observations (e.g. Argo float profiles, ship hydrography, satellite altimetry). The process of constructing a state estimate involves iteratively adjusting the initial conditions, surface forcing fields, and mixing parameters in order to reduce model-data misfit. The adjustments are carried out via the 4D-Var method, whereby adjoint sensitivity fields are used to calculate the adjustments that will decrease the model-data misfit. Below we describe some relevant features of the state estimate; we refer the reader to Forget et al. (2015a) and references therein for more details.

ECCOv4 uses a latitude-longitude-cap (LLC) grid with a nominal horizontal resolution of 1°, which corresponds to roughly 40-50 km at high latitudes and roughly 110 km at the equator. In the vertical, it uses the z* rescaled height coordinate, with 50 vertical levels ranging from 10 m to 456 m. ECCOv4 features parameterized diffusion, including simple convective adjustment, diapycnal and isopycnal diffusion, and the Gaspar-Gregoris-Lefevre mixed layer turbulence closure scheme (Gaspar and Grégoris, 1990). It also includes the bolus transport parameterization of Gent and Mcwilliams (GM, 1990). Despite the relatively coarse resolution of ECCOv4, its water mass properties are in good agreement with observations, thanks in part to the 4D-Var optimization process that iteratively adjusts the spatially varying turbulent transport coefficients (Forget et al., 2015b). As a first guess, ECCOv4 uses ERA-Interim atmospheric forcing fields, which are then adjusted to reduce model-data misfit. The buoyancy, radiative, and mass fluxes use the bulk formulae of Large and Yeager (2009). The state estimate also uses fully dynamic sea ice, with buoyancy and mass fluxes recalculated according to Losch et al. (2010). Note that this setup does not use salinity restoring at the surface.

The advantage of using this state estimate is that it provides a physically consistent description of the ocean beyond what is measured via observations, and so can be used to identify the processes behind the observed variability in the ocean. Many previous studies have used ECCOv4 to investigate variability in the North Atlantic, and have shown the state estimate to be close to be a good representation of the region. Variability in ocean heat content in the subpolar North Atlantic is well-reproduced in ECCOv4 in comparison to observations (Buckley et al., 2014; Foukal and Lozier, 2018; Asbjørnsen et al., 2019), as are trends in salinity (Tesdal et al., 2018), and overturning circulation in the subpolar North Atlantic Piecuch et al. (2017). Interannual variability in the ERA-Interim air-sea heat fluxes used to force ECCOv4 are also very similar to independent observations in the North Atlantic and Arctic Ocean (Lindsay et al., 2014). The analysis in ECCOv4-r4 set out in this paper was also repeated in ECCO Version 4 Release 3 (ECCOv4-r3, covering 1992-2015, see Forget et al., 2015a), with the same conclusions reached (Fig. A7-A17).

## 2.2 Mixed layer temperature budget

We compute mixed layer temperature budgets for the North Atlantic using a well-established analysis method (e.g. Frankignoul, 1985; Peter et al., 2006; Dong et al., 2007), as described in Equation 1 below. This method requires a number of assumptions and parameterisations to provide an approximation of the budget. We define the mixed layer depth (MLD), $h_m$, as the depth at which potential density is 0.03 kg m$^{-3}$ greater than that of the surface cell. The net rate of change in the average mixed layer temperature, $T_m$, is attributed to surface heat fluxes, horizontal advection, entrainment of water from beneath the mixed layer, vertical and horizontal diffusion, and lateral induction, which describes the horizontal transport of water through the base of a sloped mixed layer:

$$\frac{\partial T_m}{\partial t} \approx \underbrace{\frac{Q_{net} - q(h_m)}{\rho_0 c_p h_m}}_{\text{Surface flux}} - \underbrace{\mathbf{u_m} \cdot \nabla T_m}_{\text{Advection}} - \underbrace{\frac{\partial h_m}{\partial t} \frac{\Delta T}{h_m}}_{\text{Entrainment}} - \underbrace{\mathbf{u_m} \cdot \nabla h_m \frac{\Delta T}{h_m}}_{\text{Lateral induction}} + \underbrace{\frac{K_z}{h_m} \frac{\Delta T}{\Delta z}}_{\substack{\text{Vertical} \\ \text{diffusion}}} + \underbrace{\kappa \nabla^2 T_m}_{\substack{\text{Horizontal} \\ \text{diffusion}}}. \tag{1}$$

$Q_{net}$ is the net heat flux into the surface ocean, and $\rho_0$ and $c_p$ are constants denoting reference density and specific heat capacity. The decay of incoming shortwave radiation within the top depth cells is represented by the function $q$ (Chakraborty and Campin, 2013). $\boldsymbol{u}_m$ is the lateral ocean velocity averaged over the mixed layer, and $\Delta T$ is the difference between the average temperature of the mixed layer and that of the model depth cell immediately below. The entrainment velocity is defined as the rate of change in MLD, but is set to zero for a shallowing mixed layer, since detrainment does not alter the properties of the remaining water in the mixed layer.

The ECCOv4 state estimate uses a variety of diffusion schemes, with spatially-varying parameters representing diffusivity that are adjusted as part of the state estimation process. The background mixing in the model is not directly comparable to the diffusivity at the base of the mixed layer used here to compute the diffusion term of the mixed layer budget. In the budget, we use constant vertical and horizontal diffusivity values for simplicity. The values were optimized in order to minimize the average error between the right and left hand sides of the equations within the region of the cold anomaly. We therefore use a vertical diffusivity, $K_z$, of $2.01 \times 10^{-4}$ m$^2$ s$^{-1}$, and a horizontal diffusivity, $\kappa$, of 2000 m$^2$ s$^{-1}$.

This method to compute the mixed layer temperature budget results in an approximation of the budget, while taking into account the dynamic nature of the mixed layer via the inclusion of entrainment and lateral induction terms, which are driven by the spatio-temporal changes in the depth of the mixed layer. By defining the individual terms this way, we can deconstruct those relevant to the formation of the cold anomaly to further understand the processes and properties that result in anomalies in the budget terms. Using Equation 1, the mixed layer temperature budget was computed at each individual point within the approximate area covered by the North Atlantic cold anomaly, defined as 50-20°W, 43-63°N (shown in Fig. 1). For comparison, fully closed budgets were also computed online, using the output advection and diffusion terms following the steps set out by Piecuch (2017). The budget terms for each cell within the mixed layer were then averaged at each timestep to replicate the terms of the approximated mixed layer budget. The surface heat flux term of the budget is computed in the same way for each method, however the fully closed budget does not include the entrainment or lateral induction terms and so does not factor in the changing depth of the mixed layer.

## 3   Results

### 3.1   The 2015 cold anomaly in observations

We first analyse the 2015 cold anomaly in observations, for comparison with the cold anomaly in the state estimate. The average 2015 anomalies in the HadISST1 monthly averaged SST observations are shown in Fig. 1a. Negative SST anomalies are present across the majority of the North Atlantic, and are more intense when only the summer is considered (Fig. 1b). The coldest anomalies, of around -2°C, occur in the southwest of the region, centred at approximately 44°W, 48°N. The boxes shown in Fig. 1 represent the region that encompasses the majority of the cold anomaly. While the negative anomalies extend slightly further east and north, the strongest anomalies are contained within the box. In the northwest of the box, along the southwest Greenland coast, positive (warm) anomalies are only present during the summer, so do not strongly affect the 2015 annual average.

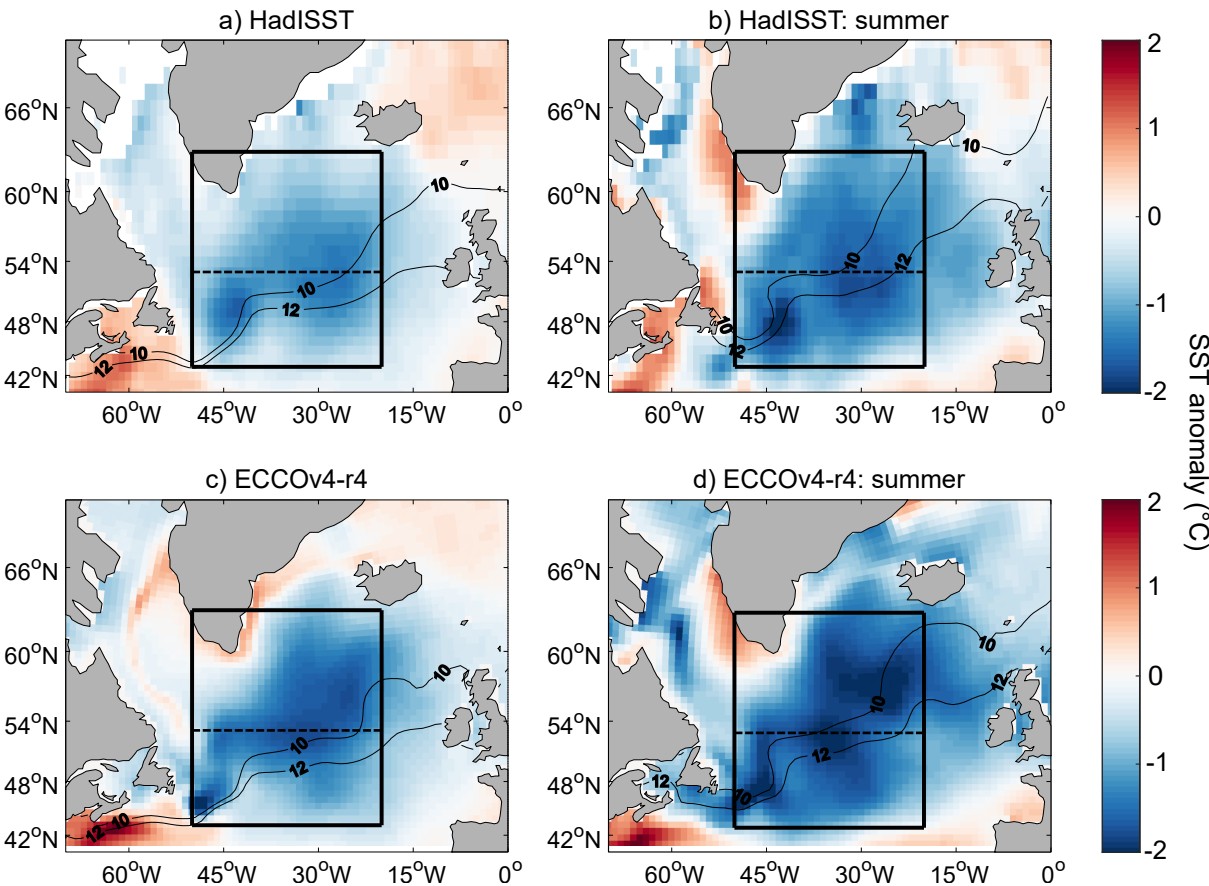

**Figure 1.** SST anomaly (°C) in a,b) the HadISST1 monthly SST observations and c,d) ECCOv4-r4, relative to the 1992-2015 climatology, averaged over the whole of 2015 (left) and the summer (JJA) only (right). The black boxes mark the region we use to define the extent of the 2015 cold anomaly (50-20°W, 43-63°N), with a dashed line at 53°N separating the north and south of the cold anomaly region. The contours show the position of the 10°C and 12°C isotherms averaged over the same period.

### 3.2 The 2015 cold anomaly in ECCOv4

The 2015 cold anomaly is present in the SST of ECCOv4-r4 (Fig. 1c,d). As in the observations, the SST anomalies are most strongly negative when only the summer of 2015 is considered, but a clear cold anomaly is also seen when the anomalies are averaged over the whole year. The state estimate captures the overall pattern of the 2015 cold anomaly seen in the observations, especially within the box focused on throughout this study, although there are some spatial differences. The negative anomaly in the 2015 average is similar to that of observations within the box, except for slight positive anomalies along the southern coast of Greenland. Positive anomalies also occur in the Labrador Sea when the SST anomaly is averaged over the whole of 2015 (Fig. 1c). However, in both the model and observations, the sign of the anomalies in this region is strongly dependent on the period over which the subtracted climatology is calculated. The positive anomalies in ECCOv4-r4 are driven by warming

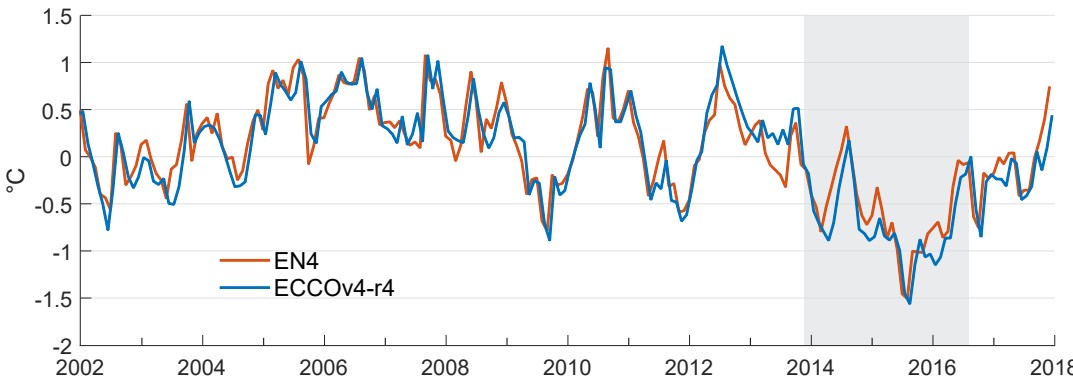

**Figure 2.** Time series of anomalies in the mixed layer average potential temperature (°C), relative to the 1992-2015 monthly climatology, averaged over the cold blob region, for EN4 observations (red) and ECCOv4-r4 (blue). The shaded area marks the time period from when the initial cold anomaly begins to emerge to when the anomaly once again becomes positive.

from January to April, but when only summer is considered, the anomalies in this region are much closer to the observations
(Fig. 1d). The position of the 10°C and 12°C isotherms is also shown in Fig. 1; strong similarities between the model and observations suggest that ECCOv4-r4 does a good job of reproducing the surface currents within the region.

In this work, we focus only on the anomalies within the box shown. Because the anomaly has no regular shape, and to remove the effect of the warm anomalies along the Greenland coast are not seen in observations, we define the cold blob region as the area within the selected control volume with an average 2015 SST anomaly below zero. The results of the mixed layer
temperature budgets are insensitive to the inclusion of these areas. When the SST anomaly is averaged over this region, the $R^2$ value between the time series of anomalies in HadISST1 observations and the state estimate is 0.94 for the period 1992-2015.

Here, we focus on the cold anomaly within the mixed layer. The time series of the mixed layer temperature anomalies averaged over the cold blob region is therefore shown for both ECCOv4-r4 and EN4 observations in Fig. 2. The 2015 cold anomaly is clear in the mixed layer temperature as the most negative anomalies over each time series, and there is good agree-
175 ment between the state estimate and the observations ($R^2 = 0.89$). Negative temperature anomalies first appear in November 2013, decreasing to -0.8°C in March 2014 in the observations, and to -0.9°C in April 2014 in ECCOv4-r4, before switching to positive anomalies during the summer. The anomalies then become negative again and decrease strongly, reaching a minimum of -1.5°C in the observations and -1.6°C in ECCOv4-r4, during August 2015. The linear trend in the anomalies in mixed layer temperature from the start of the cooling (December 2013) to the peak cold anomaly (August 2015) is -0.48°C yr$^{-1}$ in
ECCOv4-r4, compared to -0.55°C yr$^{-1}$ in the observations. The anomalies then remain predominantly negative throughout 2016 and 2017. The linear trend in the anomalies from the peak of the anomaly until the end of the ECCOv4-r4 time series (December 2017) is 0.56°C yr$^{-1}$ in ECCOv4-r4, compared to 0.53°C yr$^{-1}$ in the observations.

There are clear signatures of re-emergence in the vertical structure of the temperature anomalies and the MLD (Fig. 3). The anomaly begins in the winter of 2013/14, and reaches a minimum within the shallow summer 2015 mixed layer. During
2013, weak warm temperature anomalies extend through the water column, before cold anomalies develop in the winter of

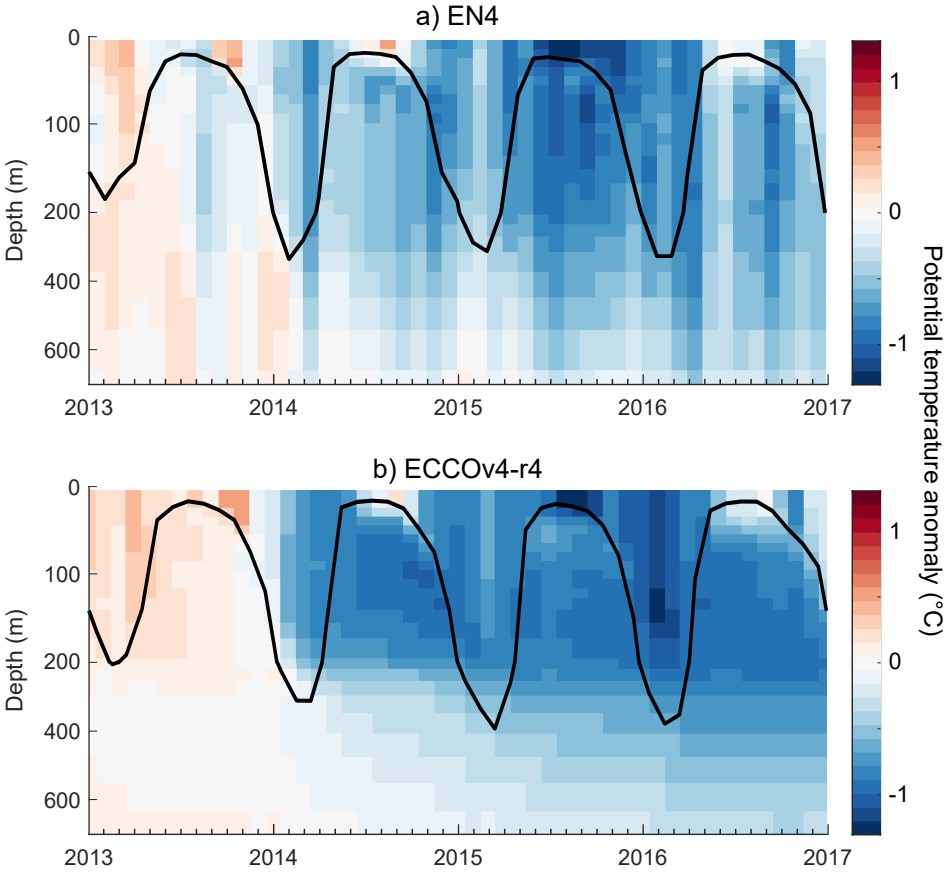

**Figure 3.** Potential temperature anomaly over depth, relative to the 1992-2015 monthly climatology (color; °C), averaged over the cold blob region in a) EN4 observations and b) ECCOv4-r4. The MLD is also shown (black line, m). Note the non-uniform spacing of the vertical axes.

2013/14, extending throughout the deep winter mixed layer. The following summer, the mixed layer shallows and the negative anomalies are sequestered beneath, where they continue to decrease slowly. During this time, the mixed layer temperature instead experiences positive (warm) anomalies, as seen within the average mixed layer temperature in Fig. 2. In October 2014, the mixed layer starts to deepen again and the cold anomaly re-emerges within the mixed layer. Further cooling occurs during the summer of 2015, with the minimum temperature occurring within the shallow summer mixed layer. Following the minimum, the cold anomaly is sustained through 2016 at a lesser magnitude, and is again sequestered below the mixed layer during the summer when very small anomalies are seen within the shallow mixed layer. In general, both the warm anomalies prior to the formation of the cold anomaly, and the cold anomaly itself, are more intense in ECCOv4-r4 than in the EN4 observations. However, there is still very little difference between the magnitude of the two sets of anomalies when the average temperature of the mixed layer is considered (Fig. 2).

At its deepest, the cold anomaly extends from the surface to depths of at least 500 m. During the formation of the cold anomaly, the depth of the winter mixed layer within the cold blob region also increases. The winter mixed layer is anomalously

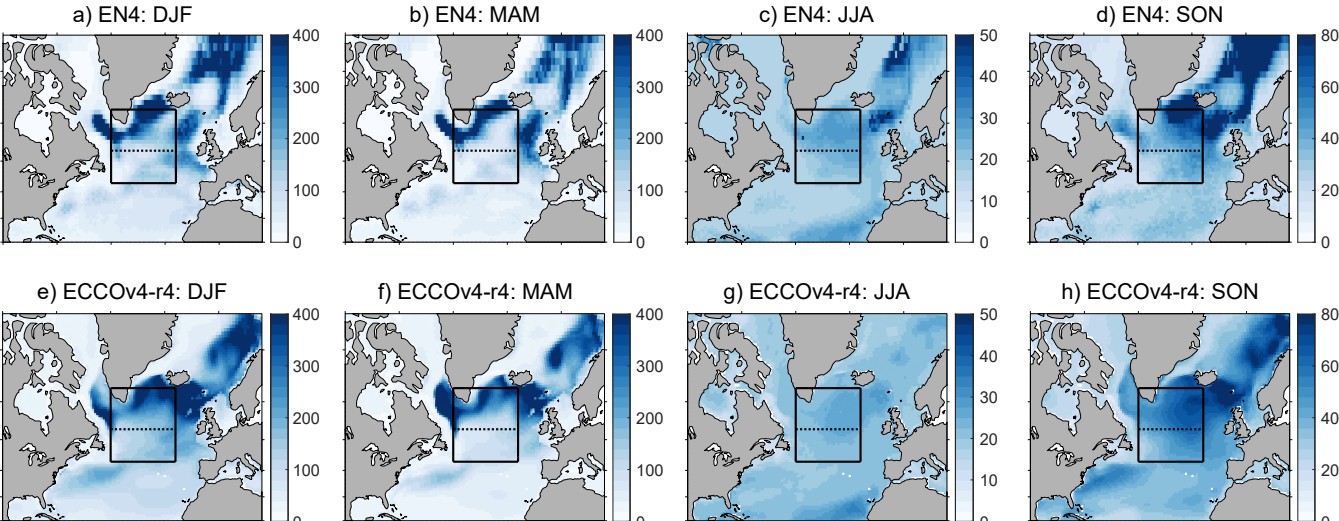

**Figure 4.** The average seasonal cycle of the North Atlantic MLD (m) from 1992-2015 in the EN4 observations (top row) and ECCOv4-r4 (bottom row). Note the varying color scales between seasons. The boxes show the position of the cold blob region, with the dashed line separating the northern and southern sections.

deep in each of the three years shown, with a maximum of 335 m in 2014 in the observations. In the state estimate, MLD instead peaks in 2015, with a maximum depth in of 393 m during March. This increase in MLD is likely a result of the the cold anomaly preconditioning the water column for deep convection, as suggested by Piron et al. (2017). To ensure ECCOv4-r4 accurately represents MLD across the North Atlantic, the seasonal cycle in MLD is compared with that calculated from EN4 temperature and salinity observations (Fig. 4). Analysis of global mixed layer depths has also been completed by Forget et al. (2015a). Within the cold anomaly region, patterns in MLD are fairly consistent between the state estimate and observations, particularly during winter and spring when mixed layers are deepest. Mixed layers are particularly deep in the northern half of the cold anomaly region, along the south coast of Greenland, and in the Irminger Sea, in both the observations and state estimate.

## 3.3 Processes driving seasonal temperature variability in the cold blob region

To determine the processes controlling temperature variability within the cold blob region, the average seasonal cycle of the mixed layer temperature budget was calculated (Fig. 5). Our approximation of the budget is shown in Fig. 5a. This approach to computing the mixed layer temperature budget takes into account the spatio-temporal variability in MLD, but due to the low temporal resolution of the model data and the various assumptions made in the method, there is a residual between the actual temperature tendency of the mixed layer and the sum of the budget terms driving that tendency. The warming of the mixed layer during summer is slightly overestimated (i.e. the sum of the temperature budget terms is greater than the actual temperature tendency within the model), and the cooling during winter is also overestimated. The most likely source of error

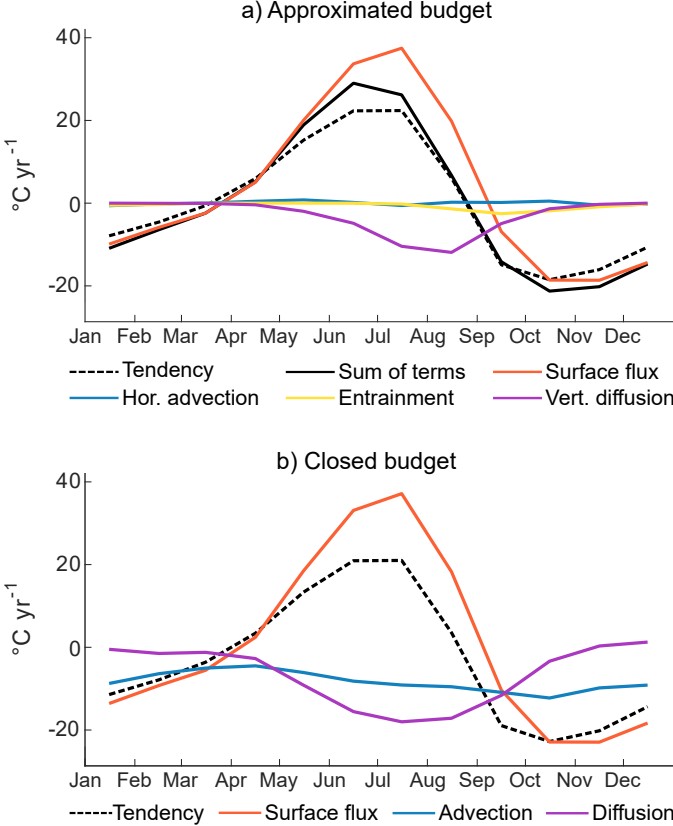

**Figure 5.** The average seasonal cycle of the dominant mixed layer temperature budget terms (°C yr$^{-1}$) averaged over the cold blob region for ECCOv4-r4, where positive values represent an increase in the rate of change in the temperature of the mixed layer. a) The approximated budget computed using Equation 1, where the black dashed line is the actual temperature tendency in the model, and the solid line is the sum of the budget terms driving that temperature change. The remaining lines represent the temperature change due to each individual process: surface heat fluxes (orange), horizontal advection (blue), vertical entrainment (yellow), and vertical diffusion (purple). Horizontal diffusion and lateral induction are not shown as the effects of both are negligible. b) The closed budget computed online for comparison, where the dashed line is the tendency, which is equal to the sum of surface fluxes (orange), advection (blue), and diffusion (purple).

during spring/summer is in the diffusion and advection terms, as entrainment should be low during this period, and surface fluxes are computed in the same way as for the closed budget and so correct for the model. However, the majority of the seasonal temperature variability is captured by the budget. The horizontal diffusion and lateral induction terms are not shown as they have a negligible effect on temperature variability in the cold blob region.

The average seasonal cycle in temperature tendency within the region is dominated by the surface heat flux, which drives a
warming of the mixed layer from April to August and a cooling of the mixed layer from September to March. The maximum seasonal warming due to surface forcing occurs in July (37.5°C yr$^{-1}$) and the maximum cooling occurs in November (-18.6°C yr$^{-1}$). Vertical diffusion is the second most important term, driving a cooling of the mixed layer from approximately May to

September, with the maximum cooling occurring from July to August, and reaching -11.9°C yr$^{-1}$. This seasonal variability in diffusive cooling is driven by the seasonality in the temperature difference between the mixed layer and thermocline, with a greater difference when the mixed layer is shallow in summer. This relationship is further discussed in section 3.4. No vertical entrainment occurs from December to July due to the shallowing mixed layer, but entrainment cools the mixed layer from August to November, as the mixed layer deepens and entrains colder water from below. The maximum mixed layer cooling via entrainment occurs in September, when the rate of deepening of the mixed layer is highest. However, the impact is small, causing a maximum cooling of -2.5°C yr$^{-1}$. Advection is also low throughout the year. When comparing with the seasonal cycle in the fully closed budget (Fig. 5b), the approximated budget does not reproduce the warming via diffusion from October to December. This is because the approximated vertical diffusion is always negative due to a negative temperature gradient across the base of the mixed layer, and the approximated horizontal diffusion is negligible.

### 3.4    Processes driving the 2015 cold anomaly

The 2015 cold anomaly is driven by a combination of surface forcing, vertical diffusion and entrainment, as shown by monthly anomalies in the temperature budget (Fig. 6). Anomalies in these processes are in turn the result of anomalies in the net surface heat flux, the temperature gradient across the base of the mixed layer, and the depth of the mixed layer (Fig. 7). The mixed layer budget prior to the removal of the seasonal cycle is also shown for the same period in the Appendix (Fig. A1) to clarify the actual sign of each term. Initial cooling of the mixed layer in the winter of 2013/14 is due to strong negative anomalies in the surface heat flux term, signifying stronger than average surface heat loss. The strongest surface-driven cooling of the mixed layer is in December 2013, with anomalies of -8.4°C yr$^{-1}$. While the change in temperature due to surface fluxes is dependent on both the net heat flux through the surface and the MLD (see Equation 1), it is anomalies in the net surface flux that dominate anomalies in the mixed layer temperature change due to surface fluxes, as can be seen by the strong correlation between the terms in Fig. 7a. Strong anomalies in the outgoing flux during December cause the stronger mixed layer cooling during this period, with anomalies of at least -100 W m$^{-2}$. This cooling is enhanced by anomalies in advective cooling of the mixed layer, reaching -3.3°C yr$^{-1}$. Advective cooling also peaks in December 2013 and is due to a combination of both zonal and meridional advection. Approximately 75% of the anomalous initial cooling of the mixed layer is therefore due to surface forcing, while the remaining 25% is predominantly a result of horizontal advection.

Following the initial cooling, anomalies in each of the main budget terms are then low until April 2014, when higher than average surface warming drives overall positive anomalies in the temperature tendency from April to June (Fig. 6). Since the mixed layer is shallow over this period, while the surface warming acts to increase the average temperature of the mixed layer, it has a relatively small effect on the heat content of the water column. The warming of the mixed layer is also suppressed by strong vertical diffusion, driven by the high temperature difference between the mixed layer and thermocline (Fig. 7c). While there is a significant temperature difference across the base of the mixed layer during the winter of 2014, due to the sequestration of the cold anomaly and further cooling beneath the mixed layer, this does not lead to diffusive cooling due to the anomalously deep mixed layer masking the impact (see the diffusion term of Equation 1). The temperature difference is greatest during summer when the mixed layer is shallow and heated by strong surface heat fluxes. The temperature difference then rises

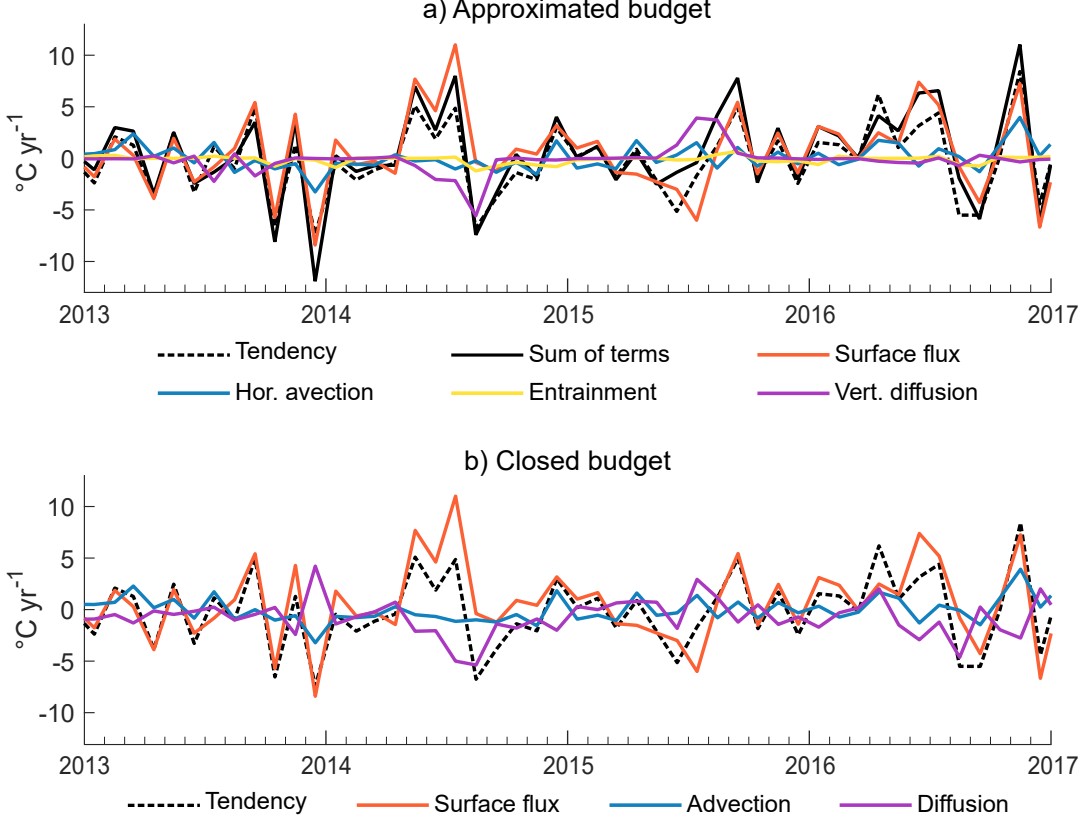

**Figure 6.** a) Anomalies in the dominant terms of the mixed layer temperature budget, relative to the 1992-2015 monthly climatology (°C yr$^{-1}$), averaged over the cold blob region. a) The approximated budget computed via Equation 1, where the dashed black line shows anomalies in the model temperature tendency and the solid black line shows anomalies in the sum of the temperature budget terms driving the temperature change. The remaining lines represent those individual processes: the surface heat flux (orange), horizontal advection (blue), vertical entrainment (yellow), and vertical diffusion (purple). The budget is also shown in Fig. A1, prior to the removal of the seasonal cycle. b) Anomalies in the closed budget for comparison.

from May onwards, driving cooling of the mixed layer via diffusion. The temperature difference peaks in August 2014 with a difference of 1.4°C, driving anomalies in diffusive cooling of -5.6°C yr$^{-1}$. This anomalously strong vertical diffusion is the main cause of the re-emergence of the cold anomaly after being sequestered beneath the mixed layer during spring/summer.

The strong summer surface warming of the mixed layer then reduces quickly in August as the anomaly in the net surface heat flux becomes close to zero (Fig. 7a) and the mixed layer deepens.

The re-emergence of the cold anomaly is also enabled by the deepening of the mixed layer, which results in the entrainment of colder water from below. The mixed layer cooling via entrainment in the model is small, with anomalies peaking during August at -1.2°C yr$^{-1}$. The depth of the mixed layer is not particularly anomalous during the winter of 2014 (Fig. 7b), especially

in comparison to the following year, and the timings of the minima in entrainment anomalies do not always correspond to

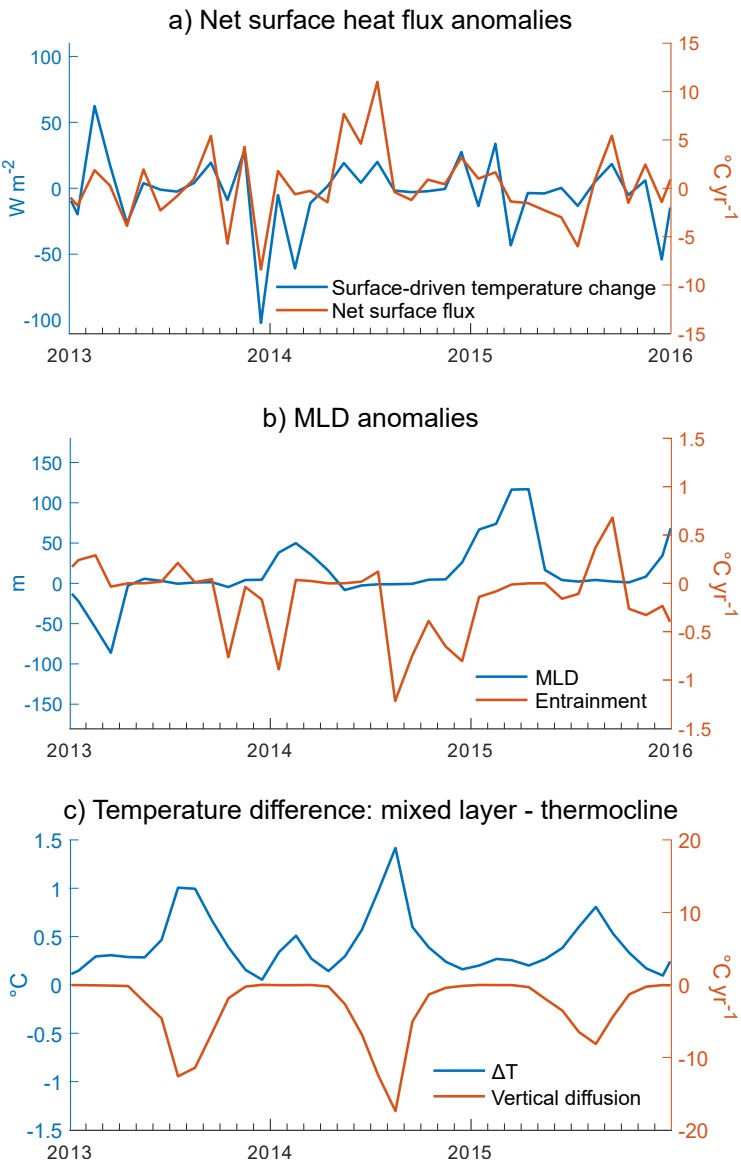

**Figure 7.** The individual components that comprise the dominant temperature budget terms averaged over the cold blob region. a) Anomalies in the net surface heat flux output by the model, defined as $Q_{net}$ in Equation 1 (W m$^{-2}$; blue), and anomalies in the associated change in mixed layer temperature, i.e. the surface flux term of the budget (°C yr$^{-1}$; red). b) Anomalies in MLD (m; blue) and the associated heat entrainment term of the budget (°C yr$^{-1}$; red). c) The temperature difference between the mixed layer and the model cell immediately beneath (°C; blue), defined as $\Delta T$ in Equation 1 where positive values signify that the mixed layer is warmer than the thermocline, and the associated vertical diffusion term of the mixed layer budget (°C yr$^{-1}$; red). Note, the seasonal cycle has not been removed in c). The time series of a) and b) prior to the removal of the seasonal cycle can be seen in Fig. A2.

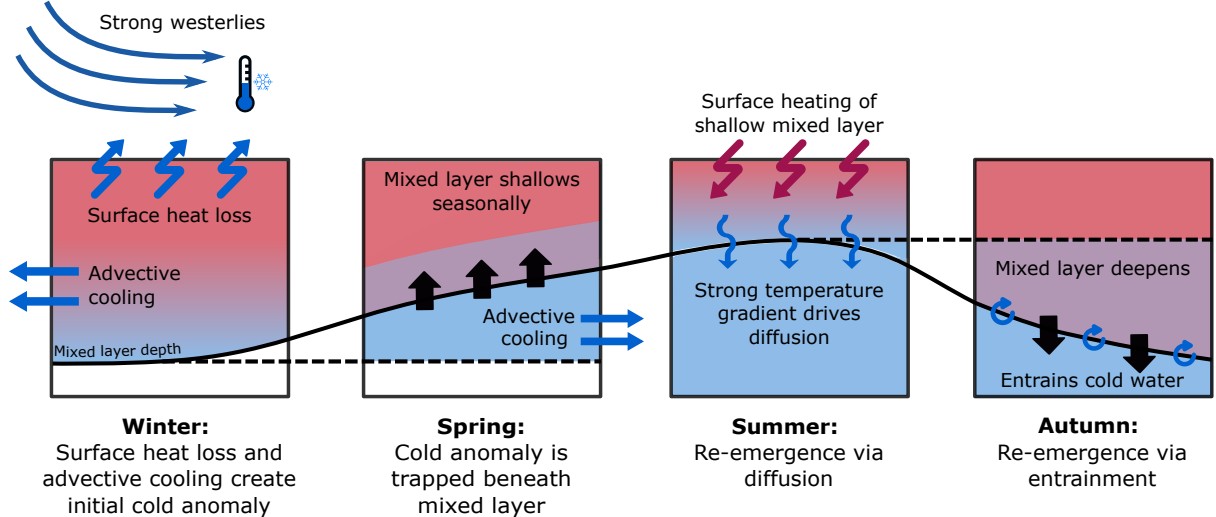

**Figure 8.** Schematic summarising the processes driving the re-emergence of the cold anomaly from beneath the mixed layer. The black arrows show the movement of the mixed layer and the red/blue arrows show the heat transfer. Initial surface heat loss and horizontal advection drive a cold anomaly in the deep mixed layer during winter, which is sequestered as the mixed layer shallows the following spring/summer. During summer, surface heating of the shallow mixed layer and potential advective cooling beneath the mixed layer generate a strong temperature gradient across the base of the mixed layer, resulting in strong diffusive cooling and driving the initial re-emergence of the cold anomaly. The mixed layer then deepens and cold water is entrained from beneath the mixed layer, driving further re-emergence.

anomalies in MLD. It is therefore the large temperature difference between the mixed layer and thermocline that drives the re-emergence of the cold anomaly, primarily through vertical diffusion, with the cooling of the mixed layer enhanced by entrainment during the autumn of 2014. The process of the re-emergence following the strong cooling of December 2013 is illustrated in Fig. 8. From June 2014 to January 2015, if the process of re-emergence is taken to be the sum of the cooling driven by vertical diffusion and entrainment, vertical diffusion is responsible for approximately 70% of the re-emergence, while entrainment is responsible for the remaining 30%.

Anomalies in each term of the mixed layer budget over the winter of 2014/15 are small. Further negative anomalies in the surface forcing then occur in the summer of 2015, signifying that, while surface heating is positive, it is around 10% lower than the average from May to July 2015, and is exacerbated by the shallow summer mixed layer. Negative anomalies in the surface flux term occur from March to July 2015, reaching a minimum of -6.0°C yr$^{-1}$ during July. This then leads to the maximum cold anomaly in August 2015, within the shallow mixed layer. Immediately after the peak of the anomaly, positive anomalies in the surface flux and diffusion terms as a result of stronger than average surface warming and weaker than average diffusive cooling, cause a decline in the intensity of the cold anomaly. However, over the winter of 2015/16, anomalies in each term of the temperature budget are small, allowing for the diminished cold anomaly to continue through to 2016. In the summer of 2016, anomalously strong mixed layer warming via surface forcing erodes the cold anomaly, with the average mixed layer temperature anomaly in the region switching to positive in August 2016.

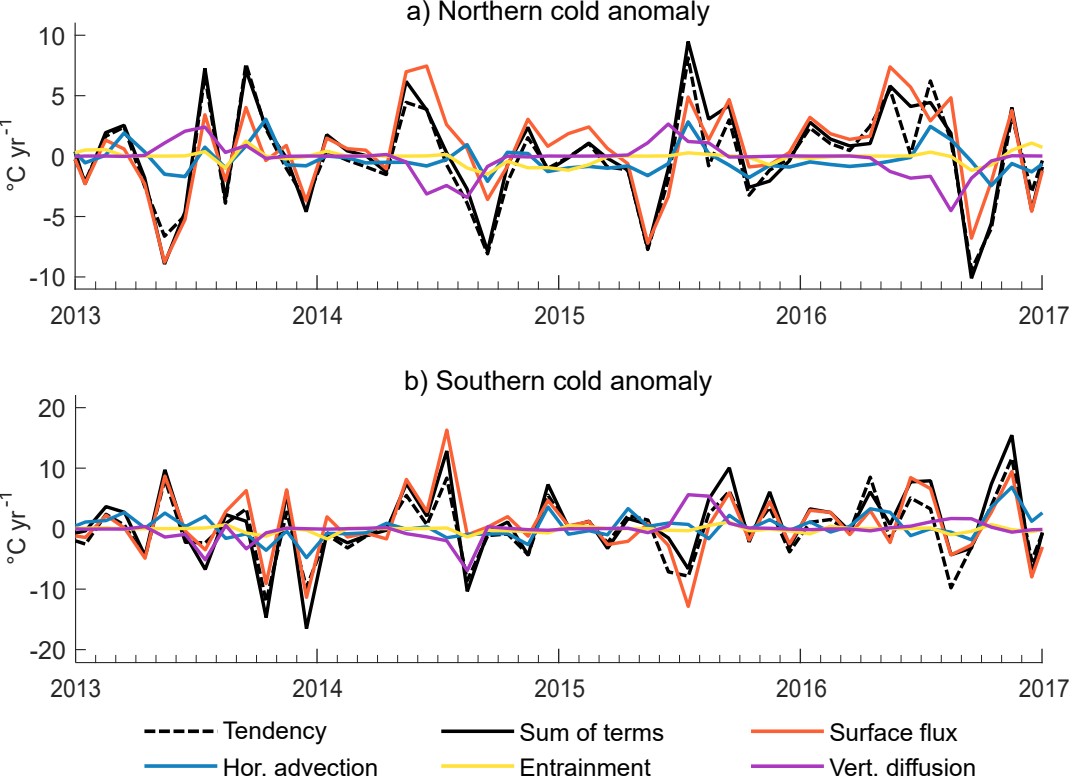

**Figure 9.** Anomalies in the mixed layer temperature budget (°C yr$^{-1}$) in ECCOv4-r4 as in Fig. 6b, but averaged over only a) the northern half of the cold blob region (50-20°W, 53-63°N) and b) the southern half of the cold anomaly (50-20°W, 43-53°N). Note the different scales on the vertical axes. The same is shown, prior to the removal of the seasonal cycle, in Fig. A4.

### 3.5 Spatial patterns in the 2015 cold anomaly

Anomalies in the temperature budget terms in the lead up to the 2015 cold anomaly are not uniform across the subpolar North Atlantic, with the most extensive differences between the north and the south of the region. The budgets were therefore repeated
and the anomalies averaged over the north and south of the cold blob region separately (Fig. 9; note the different scales on the vertical axes), with the removed climatology and the time series prior to the removal of the seasonal cycle shown in the Appendix (Fig. A3, A4). Again, the components comprising the individual budget terms are also shown, in order to further understand what causes the anomalies in the budget terms in each region (Fig. 10). In general, the magnitude of the anomalies in the temperature budget terms is reduced in the north due to a meridional gradient in winter MLD, with deeper mixed layers
in the north of the region.

The surface-driven cooling that causes the initial cold anomaly is much stronger in the south, with two clear peaks in October and December 2013 (Fig. 9b). The strongest surface-driven cooling of -11.4°C yr$^{-1}$ occurs in December, compared to -3.7°C yr$^{-1}$ in the north. This is a result of the stronger surface ocean heat loss (Fig. 10a,b) and because the mixed layer is generally

shallower in the south, so smaller anomalies in net heat flux are required to impact the average mixed layer temperature. A
second, earlier minimum in the surface-driven cooling of the mixed layer is also clear in the south in October 2013, due to a
a higher than average heat flux out of the surface ocean (Fig. 10b). In the north, 85% of the total cooling anomalies during
December 2013 is a result of surface forcing, while the remaining 15% is due to advection. In the south, the effect of advective
cooling is greater, driving approximately 30% of the initial cooling anomalies in December 2013, while the remaining 70% is
due to surface forcing.

In the summer of 2014, surface-driven warming is stronger in the south, resulting in a much greater temperature gradient
across the base of the mixed layer than in the north (Fig. 10e,f), and subsequent stronger diffusive cooling of the mixed layer.
The maximum diffusion in the south occurs in August 2014, with anomalies reaching -7.5°C yr$^{-1}$, while the maximum in the
north reaches only -3.4°C yr$^{-1}$. This diffusive cooling is followed by stronger negative tendency anomalies in the north in
September 2014, caused by a combination of surface fluxes, advection, entrainment and diffusion (Fig. 9a). The most negative
anomalies are in the surface flux term, reaching -3.6°C yr$^{-1}$, and are caused by negative anomalies in the heat flux into the
ocean, which are not seen in the south (Fig. 10a,b). Anomalies in entrainment at this time are due to the continued strong
temperature gradient across the base of the mixed layer, rather than anomalies in the MLD (Fig. 10e). Despite the anomalies
generally being of a lower magnitude in the north, the strongest entrainment is a similar level to that in the south (Fig. 9),
meaning that entrainment plays a greater role in the re-emergence of the cold anomaly in the north. Entrainment drives a mixed
layer cooling also of -1.6°C yr$^{-1}$ in August 2014 in the south, before driving a further cooling of -1.5°C yr$^{-1}$ a month later
in the north. While the temperature gradient at the mixed layer base is weaker in the north, anomalies in the MLD are much
larger (Fig. 10c,d), leading to entrainment of a similar magnitude in both regions. In the north, anomalies in processes driving
re-emergence from June 2014 to February 2015 are approximately 60% a result of vertical diffusion, and 40% entrainment. In
the south, where the impact of entrainment is lesser, the re-emergence over this period is a result of approximately 80% vertical
diffusion and 20% entrainment.

In January 2015, strong surface heat loss in the north of the cold blob region (Fig. 10a) is not replicated in the surface flux
term of the mixed layer budget, due to concurrent large anomalies in MLD. While surface forcing still drives a cooling of the
mixed layer in January 2015, anomalies in the term are slightly positive as greater surface heat loss would be required to affect
the temperature tendency of the greater volume of water in the mixed layer. The weakened surface-driven warming (negative
anomalies) in the summer of 2015 leads in the north with a peak of -7.2°C yr$^{-1}$ in May, followed by a peak in the south of
-12.4°C yr$^{-1}$ in July (Fig. 9). In both cases, the negative anomalies are due to weak negative anomalies in the net heat flux
(Fig. 10a,b) into a shallow summer mixed layer. In both the north and the south, the anomalies in each process lead to the
strongest SST anomalies in the summer of 2015 (Fig. 11). Anomalies in the surface warming of the mixed layer are positive
from January to July 2016 in the north of the cold blob region, acting to diminish the cold anomaly. Positive anomalies in the
surface flux term in the south also reach a similar magnitude, but oscillate between positive and negative. The processes driving
the cold anomaly in the north and south of the cold blob region are illustrated in Fig. 11.

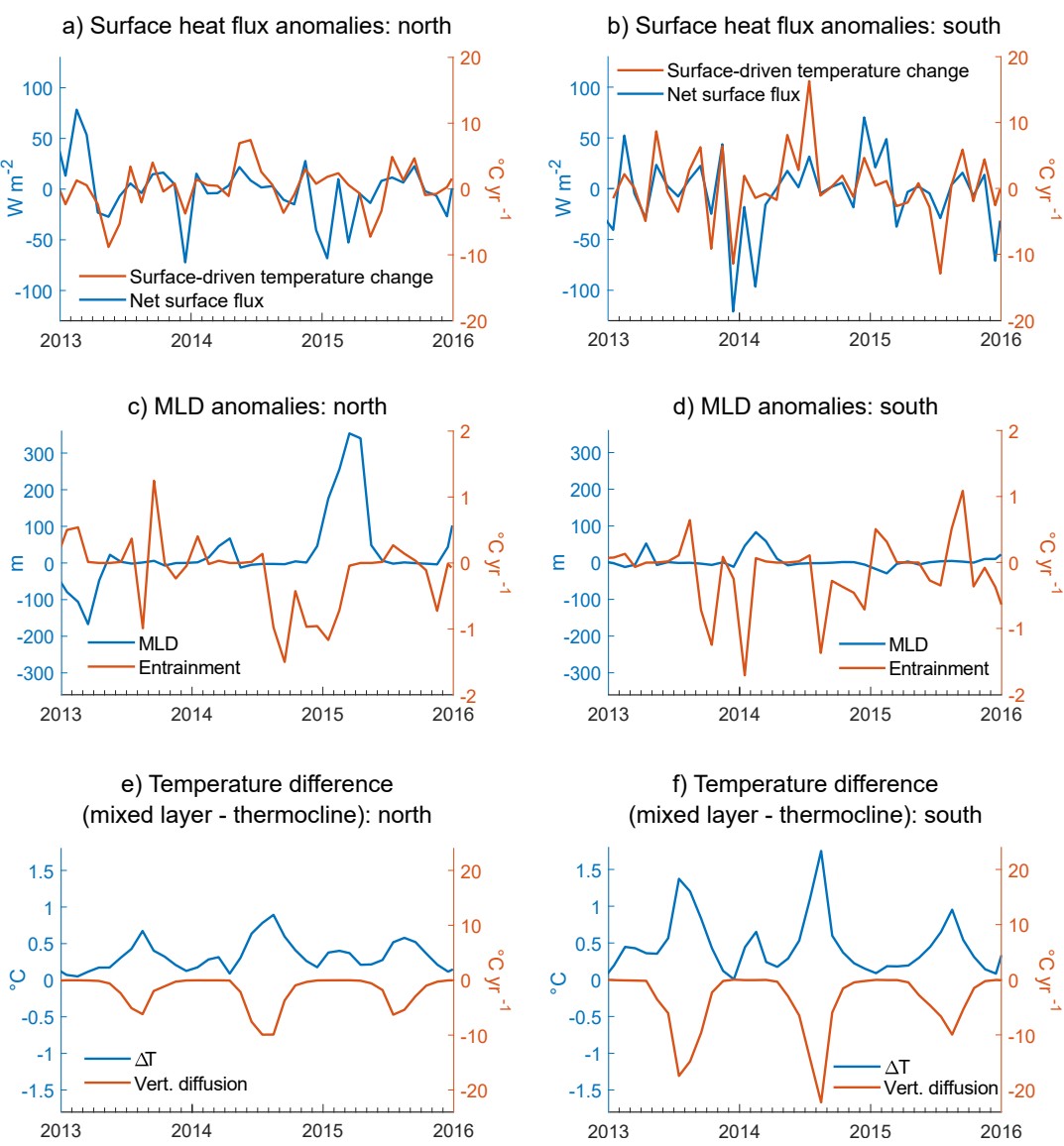

**Figure 10.** The individual components making up the dominant temperature budget terms and the associated change in mixed layer temperature, as in Fig. 7, but averaged over the north (left panels) and south (right panels) of the cold blob region separately. a-d) show anomalies in the individual terms, while the seasonal cycle has not been removed for e-f). The time series of a-d) prior to the removal of the seasonal cycle can be seen in Fig. A5.

## 3.6 Drivers of the surface-driven cooling of the mixed layer

While multiple processes are important for the evolution of the 2015 cold anomaly, the anomaly would not have developed without the initial strong surface cooling in December 2013. Since there are clear differences in the magnitude of the heat

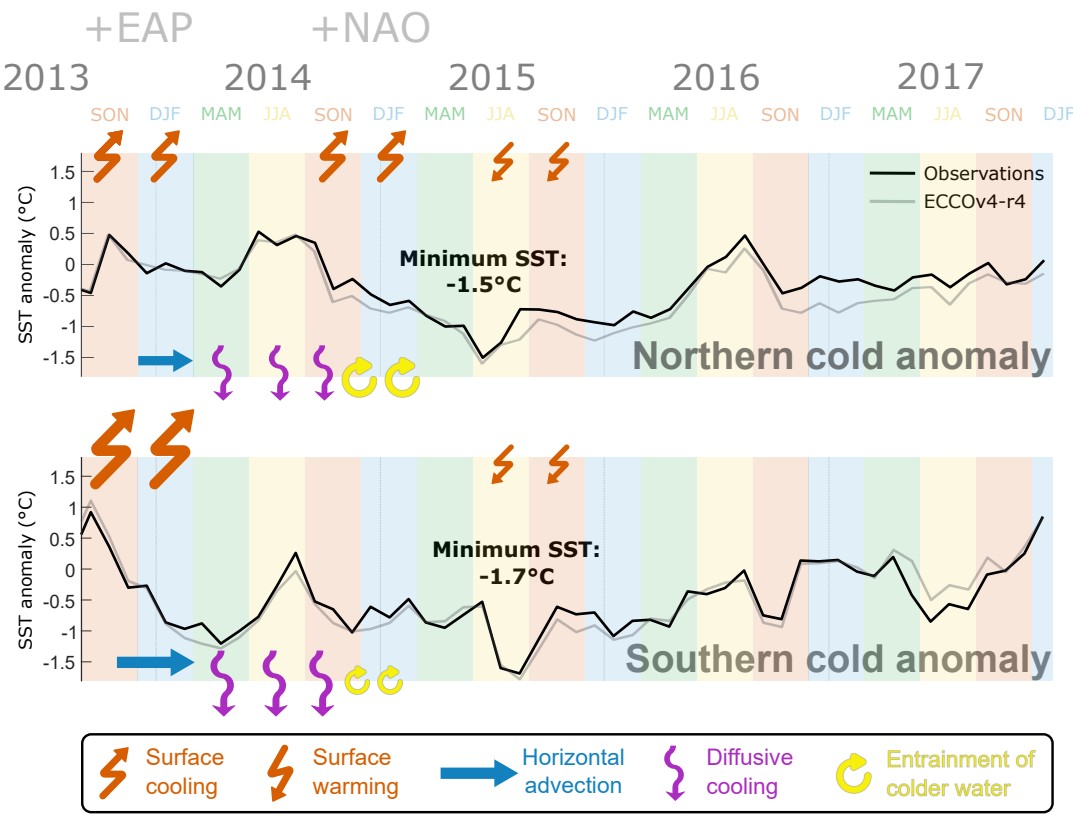

**Figure 11.** Schematic illustrating the processes involved in the evolution of the 2015 north Atlantic cold anomaly. The black line shows the SST anomaly in the HadISST1 observations averaged over the north (top) and south (bottom) of the cold blob region, with the 1992-2017 climatology removed, while the gray line shows the anomalies in ECCOv4-r4, averaged over the same region. The arrows show anomalies in the various processes driving the cold anomaly, with larger arrows representing the more important processes.

flux out of the ocean in the north and south of the cold blob region during this period (Fig. 10a,b), the spatial distribution of anomalies in that heat flux is shown in Fig. 12a. To further understand the reasons for those spatial patterns, the simultaneous anomalies in the zonal and meridional components of the surface wind stress are also shown (Fig. 12c,e) as well as the anomalies in MLD (Fig. 12g).

       While the negative anomalies in the surface heat flux extend across the majority of the subpolar North Atlantic in December

2013, the most negative anomalies occur in the cold blob region south of 54°N, and to the northwest in the Labrador Sea (Fig. 12a). Averaged over the entire cold blob region, the heat flux out of the surface in December 2013 is approximately 45% greater than the climatological mean. At the same time, the usual westerly winds over the subpolar gyre are much stronger in the southern half of the North Atlantic and in the Labrador Sea (Fig. 12c), matching the patterns of negative anomalies in the surface heat flux. The westerly winds in December 2013 are approximately twice as strong as the average December zonal wind

stress when averaged over the cold blob region. The only area of the cold blob region that does not experience anomalously

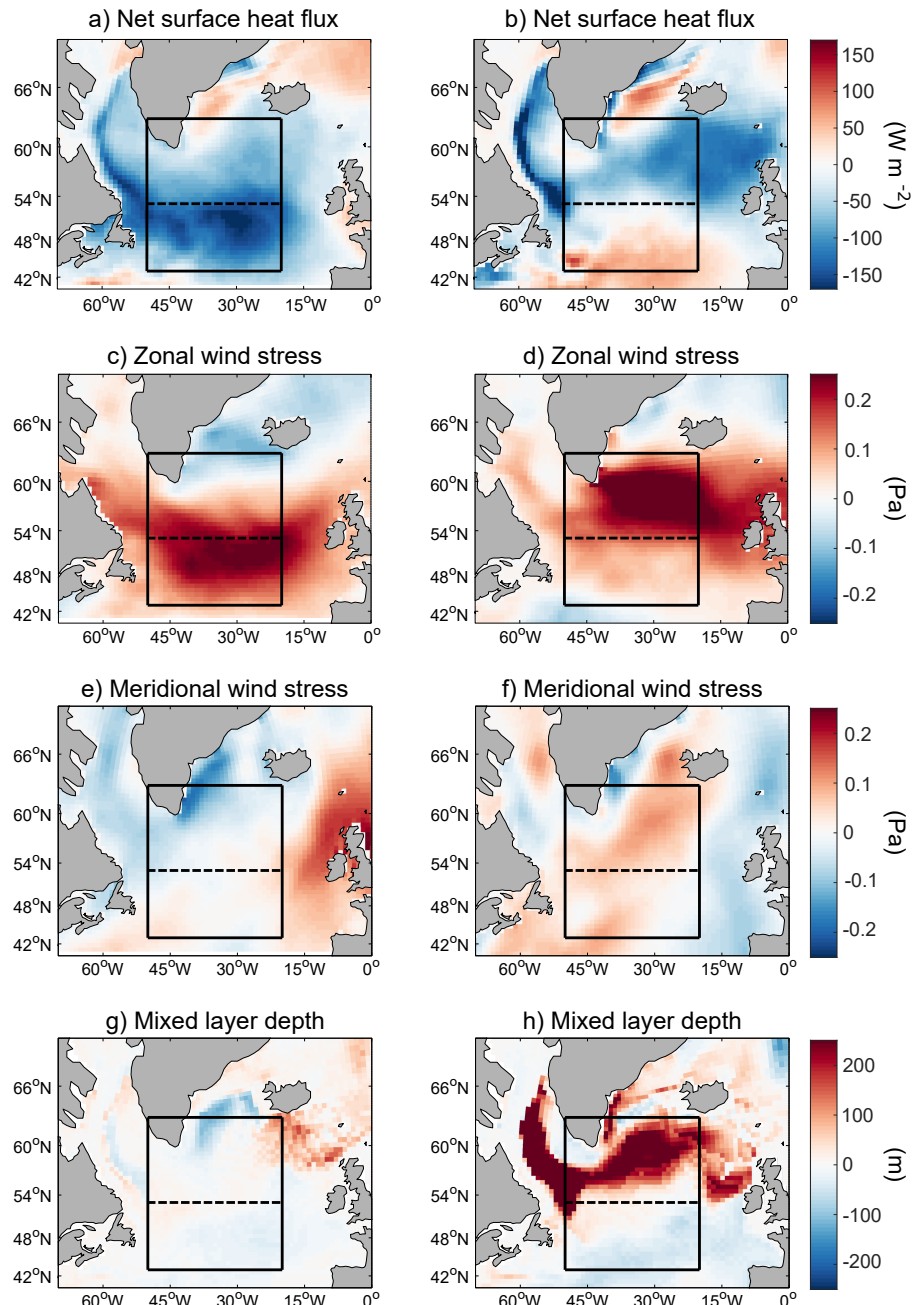

**Figure 12.** The spatial distribution of ECCOv4-r4 anomalies in the terms causing the initial anomalous surface cooling in December 2013 (left) and the same anomalies for January 2015 (right), when the net heat flux out of the surface ocean was also high but its impact not seen in the temperature of the mixed layer. Shown are anomalies in a,b) the net surface heat flux (W m$^{-2}$), c,d) zonal wind stress (Pa), e,f) meridional wind stress (Pa), and g,h) MLD (m).

strong westerly winds in December 2013 is in the far north, in the Irminger Sea. This is also the only region that experiences strong northerly wind anomalies (Fig. 12e) and positive anomalies in the net heat flux in December 2013, indicating lower than average heat loss. Anomalies in meridional wind stress are minimal across the rest of the region. Patterns in anomalies in the wind speed are also very similar to those in the zonal wind stress. These complementary patterns in surface heat flux

and wind stress anomalies suggest that the initial development of the cold anomaly is the result of anomalous local winds, either increasing air-sea heat exchange due to the increased wind speed, or via the transport of cooler air over the region. This then leads to the anomalously strong surface cooling of the mixed layer that causes the initial development of the 2015 cold anomaly.

Strong surface forcing in the winter of 2014/15 has also previously been observed, linked to the positive state of the NAO

(Yeager et al., 2016; Josey et al., 2018). However this was not seen in the anomalies of the mixed layer temperature budget, in either the north or the south of the cold blob region (Fig. 9). Strong negative anomalies in the net surface flux into the ocean were present in January 2015 and the spatial distribution of these anomalies is shown in Fig. 12b. There are clear negative anomalies in the north which have a similar spatial pattern to positive anomalies in the zonal wind stress (Fig. 12d). At the same time, anomalies in the meridional wind stress are slightly positive across the majority of the cold blob region (Fig. 12f).

Anomalies in the MLD in January 2015 (Fig. 12h) explain why the increased surface heat loss does not result in negative anomalies in the mixed layer temperature: the mixed layer is anomalously deep in the north of the cold blob region, so stronger surface forcing is required to affect the average temperature of the larger volume of water in the mixed layer. Therefore, while the mixed layer is cooling during January 2015, that cooling is no greater than the climatological average. Since the strong surface forcing extends to the east of the box defining the cold blob region (Fig. 12a,b), the mixed layer budget was repeated

for a northeastern box (35-5°W, 53-63°N; see Fig. A6). However, this had little effect on the results and the anomalies in the surface flux term of the mixed layer budget were close to zero. While anomalies in the surface heat loss were indeed greater for this region, anomalies in MLD were still large enough to mask the effect on the temperature tendency of the mixed layer.

## 4    Discussion

### 4.1    The influence of climate modes on the 2015 cold anomaly

Previous studies have highlighted the influence of two climate modes of variability, namely the NAO and the EAP, on the development of the 2015 cold anomaly (Yeager et al., 2016; Josey et al., 2018; Maroon et al., 2021). During the initial cooling in the winter of 2013/14, the EAP was the dominant climate mode in the region and at its most positive state in at least six decades (Josey et al., 2018). The patterns in wind stress in December 2013 match composites of wind speeds for winters with a positive EAP index (Josey et al., 2019), with strong westerlies across the south of the region driving the anomalously strong

heat flux out of the surface ocean. The anomalous northerly winds in the Irminger Sea during the same period also relate to the pattern of the positive EAP. This region has previously been shown to experience lower air-sea temperature and humidity gradients generated by northerly winds when the EAP index is positive (Josey et al., 2019), explaining the slightly increased surface heat loss, seen in December 2013.

During the following winter of 2014/15, the NAO was the dominant climate mode and anomalously positive. During a positive NAO event, stronger surface cooling is generally observed in the North Atlantic north of 45°, with weaker surface cooling in the south (Marshall et al., 2001). Positive NAO conditions have previously been shown to result in a particularly strong increase in the westerly winds in the Irminger Sea, due to the interaction between the large scale flow and the Greenland topography (Doyle and Shapiro, 1999; Moore, 2003), as seen in the spatial distribution of zonal wind stress anomalies in January 2015. These strong westerly winds have been linked to increased surface heat loss in the north of the cold blob region (Josey et al., 2019). While the surface heat flux out of the ocean was stronger in the north during January 2015 when the the NAO was positive, this was not reflected in the anomalies of the mixed layer temperature. The effect was masked by anomalously deep mixed layers for the simple reason that larger volumes of water do not cool as readily as smaller volumes of water. When a more northeasterly region was considered, where surface heat loss in the winter of 2014/15 was more intense, the anomalies in MLD are still great enough to largely mask the effect on the anomalies in the surface flux term of the mixed layer budget. These results suggest that while the NAO clearly drove strong anomalies in winds and surface heat loss over the subpolar North Atlantic during the 2015 cold anomaly, the anomalously strong EAP appears to have had the largest effect on the temperature of the mixed layer as a whole.

### 4.2 The re-emergence of the cold anomaly

After appearing in late 2014, the cold SST anomaly was sustained through to 2015 via the re-emergence of the cold subsurface anomaly from below the mixed layer in the summer/autumn of 2015. Previous studies have linked the re-emergence of temperature anomalies in the North Atlantic to the deepening of the winter mixed layer (Cassou et al., 2007; Taws et al., 2011). While the deepening mixed layer did result in the entrainment of colder water from below, we found that this entrainment of colder water was not enough to explain the re-emergence of the cold SST anomaly. Instead, vertical diffusion dominated, while entrainment was still important but had a weaker influence. The re-emergence, via anomalies in both vertical diffusion and entrainment, appears to have been largely a result of the strong temperature gradient across the base of the mixed layer, which was particularly high in the summer/autumn of 2014 due to summer surface warming. The relative importance of entrainment was greater in the north of the cold blob region, where deeper winter mixed layers resulted in larger entrainment velocities in the autumn, though it was still a secondary process in comparison to the influence of vertical diffusion.

Since the mixed layer budgets were approximated and the diffusivity values chosen in order to reduce the error in the budgets, there is some error in the magnitude of the vertical diffusion term. However, the closed mixed layer temperature budget shows similar levels of diffusive cooling during the summer/autumn of 2014 (Fig. 6b), giving further confidence in our results. Our chosen method of computing the budgets allows us to directly relate the levels of entrainment and vertical diffusion to changes in MLD and temperature, in order to describe the process of the re-emergence of the cold anomaly in greater detail.

### 4.3 The influence of advection on the cold anomaly

While previous studies have shown the importance of advection in driving variability in the upper ocean heat content of the North Atlantic (Buckley et al., 2014, 2015), they are not directly comparable with ours as they adopted a climatological monthly

depth for the upper ocean, rather than considering interannual variability in MLD. We found that the effect of advection on the temperature of the cold blob region as a whole was small on seasonal timescales, relative to the effect of surface forcing during the cold anomaly event. However, on interannual timescales advection played a larger role. Advection drove approximately a quarter of the initial cooling of the mixed layer during the winter of 2013/14, and was therefore important in causing the cold anomaly to reach the magnitude it did.

The results of the mixed layer budgets averaged over the entire cold blob region only show the net impact of advection transporting heat in and out of the region and not the redistribution of heat within it. However, when the northern and southern halves of the cold blob region were considered separately, advection still played only a smaller role in the initial cooling in comparison to surface forcing. The fact that the cold anomaly continued to cool while sequestered beneath the mixed layer in the spring/summer of 2014 and 2015 suggests that advection beneath the mixed layer was important in increasing the magnitude of the anomaly. This cooling below the mixed layer then acted to further increase the temperature gradient across the base of the mixed layer, enhancing the cooling of the mixed layer via vertical diffusion. The stronger role of advection in the south of the cold blob region may confirm the findings of Holliday et al. (2020), with respect to changes in circulation of the period of cooling and freshening in the North Atlantic. From 2014–2016, they found that the Labrador Current flowed primarily into the North Atlantic Current zone, driving a fresh anomaly. These changes are likely to have resulted in temperature anomalies as well as the observed freshening.

Since advection played a smaller role in comparison to surface forcing in driving the cooling that caused the cold anomaly to develop, particularly in the north of the region, we can conclude that the 2015 cold anomaly was largely the result of vertical processes, i.e. surface forcing, vertical diffusion, and entrainment. The dynamics of the 2015 cold anomaly could therefore potentially be represented by a one-dimensional model, albeit with an underestimated magnitude due to the lack of initial advective cooling.

### 4.4 Increased convection during the cold anomaly

During the 2015 cold anomaly, the depth of the winter mixed layer in the cold blob region increased, reaching a maximum depth during the winter/spring of 2015. At the same time, the mixed layer in the region was undergoing a longer term freshening (Holliday et al., 2020), which is also present in the ECCOv4 state estimate (Fig. 13). The freshening indicates that the interannual deepening of the winter mixed layer was the result of stronger temperature-driven convection, while changes in salinity instead acted to stratify the mixed layer. The subsequent increased production of Subpolar Mode Water following the enhanced convection has also been suggested to have exacerbated the freshening, via its impact on the velocity of the North Atlantic Current (Holliday et al., 2020). The timing of the fresh anomalies in ECCOv4 support this theory, with the strongest fresh anomalies occurring in the aftermath of the peak cold anomaly and the enhanced convection.

Average winter mixed layers are deeper in the north of the North Atlantic in comparison to further south. The MLD anomalies relative to the climatology were also much deeper in the north of the cold blob region over the evolution of the cold anomaly. MLDs were especially deep in the winter of 2015, explaining the stronger influence of entrainment in the north. The increased convection also had subsequent effects on the other terms of the temperature budget, with the differences in the depth of

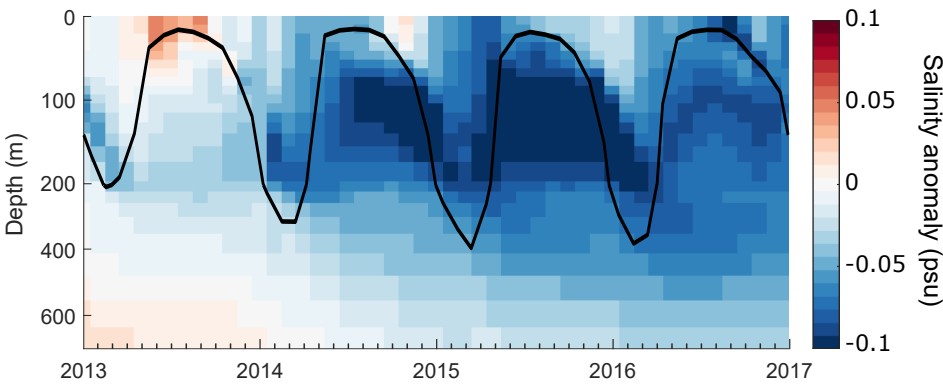

**Figure 13.** Salinity anomalies over depth relative to the 1992-2015 monthly climatology (color; psu), averaged over the cold blob region in ECCOv4-r4 for the period leading up to and during the 2015 cold anomaly. The MLD is also shown (black line, m). Note the non-uniform spacing of the vertical axis.

the mixed layers between the northern and southern regions leading to differences in how the mixed layer was impacted by anomalies in the overlying atmospheric conditions. This shows the importance of considering the spatial patterns in the drivers of this and similar cold anomalies in the North Atlantic, particularly the meridional differences in forcing.

### 4.5 Caveats of using ECCOv4

While there are many benefits of using state estimates such as ECCOv4 to investigate ocean variability, particularly when computing budgets, there are also caveats that should be taken into account. The spatial resolution of ECCOv4 is relatively coarse, which could lead to bias due to the lack of explicitly resolved sub-grid scale processes. For example, the coarse resolution of the model could result in an underestimation of eddy heat fluxes, a process known to be important in this region from observations (Foukal and Lozier, 2018). However, the mixing parameters in the model have been optimized via the state

estimation process to better represent these unresolved processes, partially offsetting the limitations of the coarse resolution (Forget et al., 2015b)

State estimates are constrained by a set of available observations; some of these observations are relatively uncertain and therefore only offer a weak constraint for the state estimation process. For example, estimates of surface heat flux can be particularly uncertain (Grist and Josey, 2003). However, the reanalysis air-sea heat fluxes used to force ECCOv4 are consistent

with independent observations North Atlantic and Arctic Ocean (Lindsay et al., 2014).

## 5 Conclusions

In this work, we used the ECCOv4 state estimate to analyse the causes of the 2015 North Atlantic cold anomaly. The anomaly was primarily driven by strong surface forcing; specifically, anomalous winds were responsible for the majority of the initial cooling in the winter of 2013/14. This cooling was strongest in the south of the anomaly region, related to the strongly positive

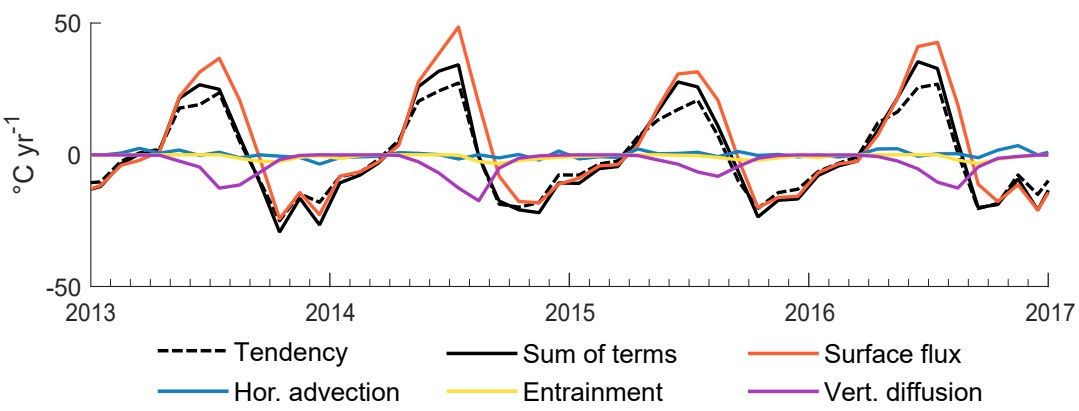

**Figure A1.** The dominant terms of the ECCOv4-r4 mixed layer temperature budget from 2013-2016 (°C yr$^{-1}$), averaged over the cold blob region. The dashed black line shows the model temperature tendency, and the solid black line shows the sum of the temperature budget terms driving the temperature change. The remaining lines represent those individual processes: the surface heat flux (orange), horizontal advection (blue), vertical entrainment (yellow), and vertical diffusion (purple). Lateral induction and horizontal diffusion are not shown because both are negligible.

EAP. The re-emergence of the cold anomaly the following winter was primarily driven by vertical diffusion due to a strong temperature gradient across the base of the mixed layer, while entrainment over the same period was relatively weak. Although the NAO was strongly positive in the winter of 2014/15, the associated anomalous surface cooling in the north of the cold anomaly region was not reflected in the mixed layer temperature, as deeper winter mixed layers masked the impact of surface cooling on temperature. Advection played a smaller but important role in the evolution of the cold anomaly, however more

work on the processes occurring beneath the mixed layer would be useful for determining whether advection was the cause of the continued cooling of the sequestered cold anomaly. Further work investigating the cold anomaly in higher resolution models would also be a welcome addition to the literature.

*Code and data availability.*   The ECCOv4-r3 data can be found at https://ecco.jpl.nasa.gov/drive/files/Version4/Release3 (downloaded June 2018), and the ECCOv4-r4 data can be found at https://ecco.jpl.nasa.gov/drive/files/Version4/Release4/ (downloaded January 2021) (ECCO

Consortium et al., 2021). ECCO version 4 is described by Forget et al. (2015a), and ECCO version 4 releases 3 and 4 are described by Fukumori et al. (2017) and ECCO Consortium et al. (2021) respectively. The HadISST observational data used to produce Fig. 1 can be found at https://www.metoffice.gov.uk/hadobs/hadisst/data/download.html, and is described by Rayner et al. (2003). EN.4.2.2 data were obtained from https://www.metoffice.gov.uk/hadobs/en4/ and are ©British Crown Copyright, Met Office, 2022, provided under a Non-Commercial Government Licence http://www.nationalarchives.gov.uk/doc/non-commercial-government-licence/version/2/.

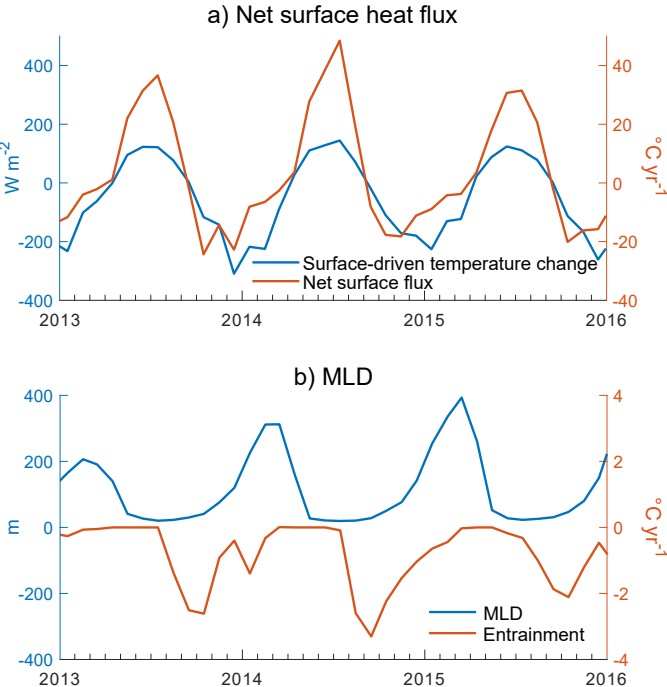

**Figure A2.** Time series of the individual components that comprise the dominant ECCOv4-r4 temperature budget terms, and the associated change in mixed layer temperature, averaged over the cold blob region. a) The heat flux into the surface output by the model, defined as $Q_{net}$ in Equation 1 (W m$^{-2}$; blue), and the associated change in mixed layer temperature, i.e. the surface flux term of the mixed layer budget (°C yr$^{-1}$; red). b) MLD (m; blue) and the associated heat entrainment term of the mixed layer budget (°C yr$^{-1}$; red).

**Appendix A**

*Author contributions.* RS completed the analysis, DJ proposed the study, DJ, SJ, BS, GF provided advice on the analysis.

*Competing interests.* No competing interests.

*Acknowledgements.* RS was supported by the UK Natural Environment Research Council ACSIS program: NE/N018028/1. DJ was supported by UK Research and Innovation grant: MR/T020822/1. SJ and BS were supported by the NERC ACSIS program (NE/N018044/1).
GF is supported by NASA award 1553749 and Simons Foundation award 549931. The authors thank An T. Nguyen and Helen Pillar for their helpful discussion in the analysis of the mixed layer budgets.

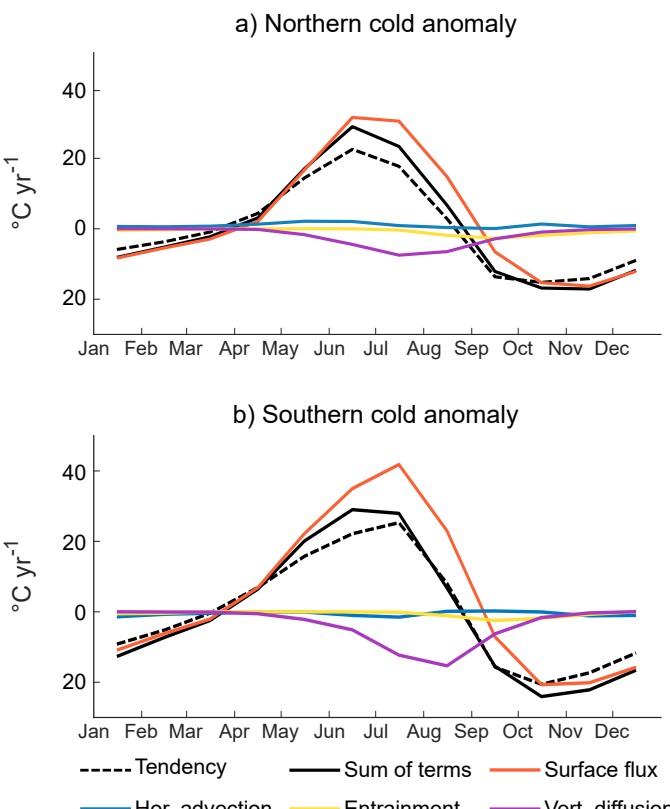

**Figure A3.** The average seasonal cycle in the dominant terms of the mixed layer temperature budget averaged over the northern (50-20°W, 53-63°N; top row) and southern (50-20°W, 43-53°N; bottom row) half of the cold anomaly region (°C yr$^{-1}$) for ECCOv4-r4. The model temperature tendency is shown by the dashed black line, and the sum of the temperature budget terms driving the change in temperature by the solid black line. The remaining lines represent those individual terms: the surface heat flux (orange), horizontal advection (blue), vertical entrainment (yellow), and vertical diffusion (purple).

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

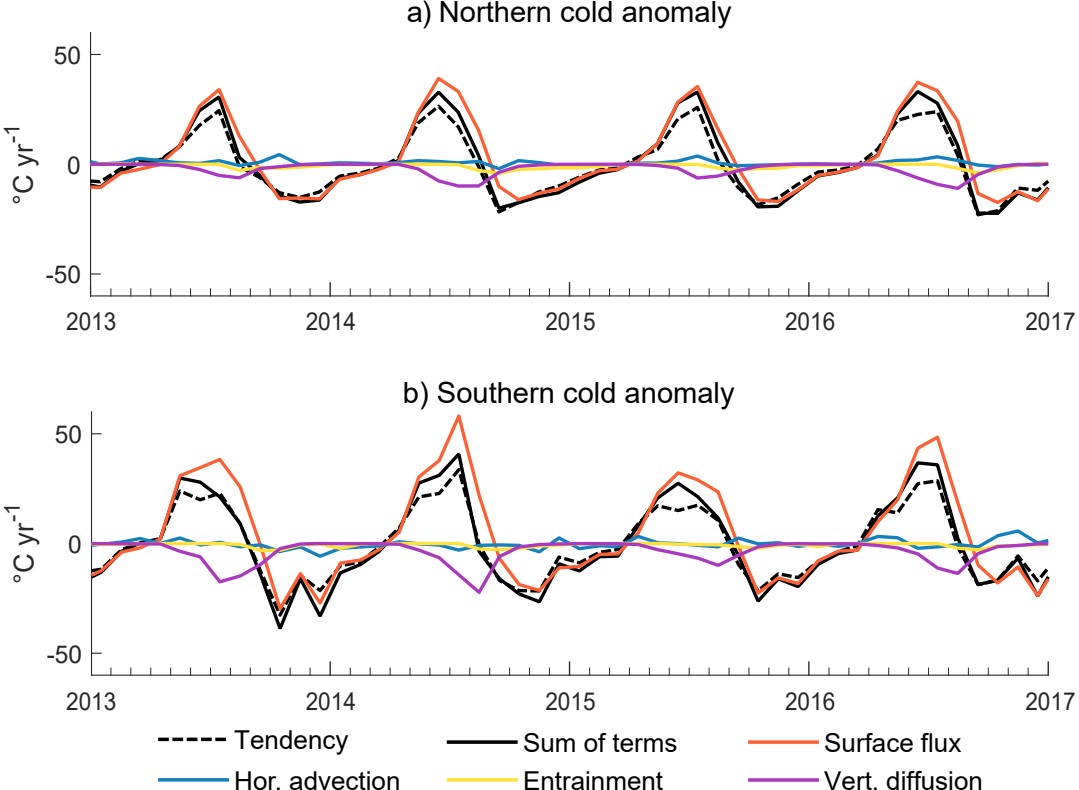

**Figure A4.** The dominant terms of the mixed layer temperature budget from 2013-2016 (°C yr$^{-1}$) in ECCOv4-r4, averaged over the a) northern and b) southern half of the cold blob region. The dashed black line shows the model temperature tendency, and the solid black line shows the sum of the temperature budget terms driving the temperature change. The remaining lines represent those individual processes: the surface heat flux (orange), horizontal advection (blue), vertical entrainment (yellow), and vertical diffusion (purple). Lateral induction and horizontal diffusion are not shown both are negligible.

Buckley, M. W., Ponte, R. M., Forget, G., and Heimbach, P.: Determining the origins of advective heat transport convergence variability in the North Atlantic, Journal of Climate, 28, 3943–3956, https://doi.org/10.1175/JCLI-D-14-00579.1, 2015.

Cassou, C., Deser, C., and Alexander, M. A.: Investigating the impact of reemerging sea surface temperature anomalies on the winter atmospheric circulation over the North Atlantic, Journal of climate, 20, 3510–3526, https://doi.org/10.1175/JCLI4202.1, 2007.

Chakraborty, A. and Campin, J.-M.: Heat and salt budgets in MITgcm, Tech. rep., Atmospheric and Oceanic Sciences Group, Space Applications Centre, Indian Space Research Organization, Ahmedabad, India, 2013.

de Jong, M. F. and de Steur, L.: Strong winter cooling over the Irminger Sea in winter 2014–2015, exceptional deep convection, and the emergence of anomalously low SST, Geophysical Research Letters, 43, 7106–7113, https://doi.org/10.1002/2016GL069596, 2016.

Desbruyères, D. G., Mercier, H., Maze, G., and Daniault, N.: Surface predictor of overturning circulation and heat content change in the
500 subpolar North Atlantic, Ocean Science, 15, 809–817, https://doi.org/10.5194/os-15-809-2019, 2019.

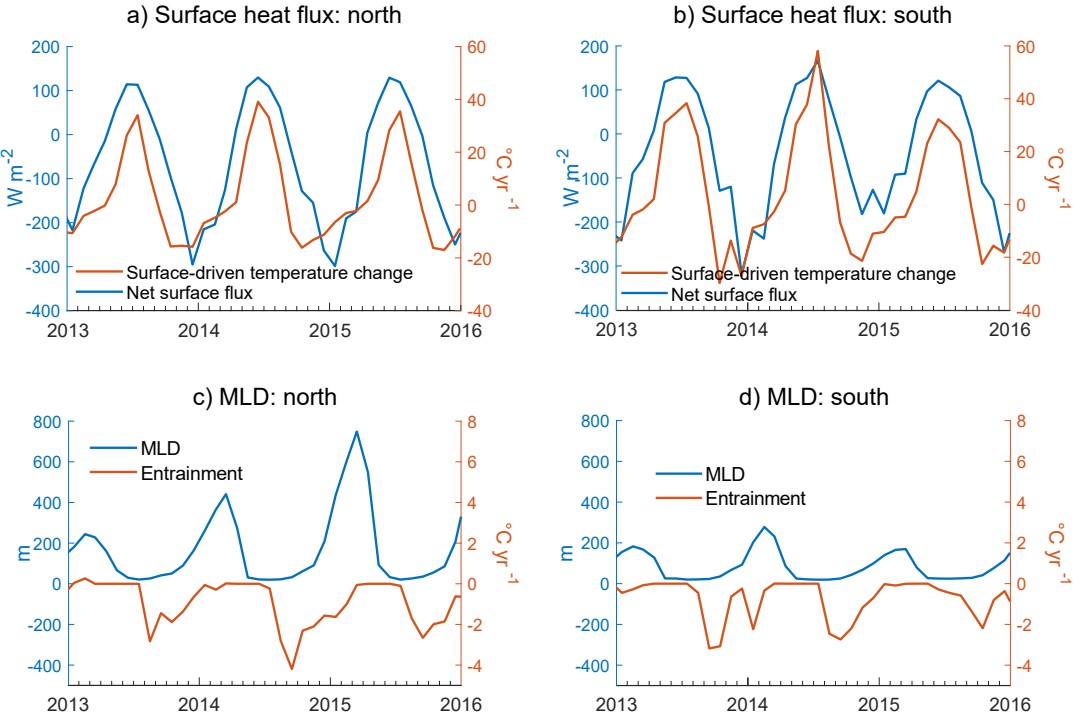

**Figure A5.** Time series of the individual components that comprise the dominant ECCOv4-r4 temperature budget terms, and the associated change in mixed layer temperature, averaged over the northern (a,c) and southern (b,d) half of the cold blob region. a) The heat flux into the surface output by the model, defined as $Q_{net}$ in Equation 1 (W m$^{-2}$; blue), and the associated change in mixed layer temperature, i.e. the surface flux term of the mixed layer budget (°C yr$^{-1}$; red). b) MLD (m; blue) and the associated heat entrainment term of the mixed layer budget (°C yr$^{-1}$; red).

Dong, S., Gille, S. T., and Sprintall, J.: An assessment of the Southern Ocean mixed layer heat budget, Journal of Climate, 20, 4425–4442, https://doi.org/10.1175/JCLI4259.1, 2007.

Doyle, J. D. and Shapiro, M. A.: Flow response to large-scale topography: The Greenland tip jet, Tellus A, 51, 728–748, https://doi.org/10.1034/j.1600-0870.1996.00014.x, 1999.

Drijfhout, S., Van Oldenborgh, G. J., and Cimatoribus, A.: Is a decline of AMOC causing the warming hole above the North Atlantic in observed and modeled warming patterns?, Journal of Climate, 25, 8373–8379, https://doi.org/10.1175/JCLI-D-12-00490.1, 2012.

Duchez, A., Frajka-Williams, E., Josey, S., Evans, D., Grist, J., Marsh, R., McCarthy, G., Sinha, B., Berry, D., and JJ-M, H.: Drivers of exceptionally cold North Atlantic Ocean temperatures and their link to the 2015 European heat wave, Environmental Research Letters, 11, https://doi.org/10.1088/1748-9326/11/7/074004, 2016.

ECCO Consortium, Fukumori, I., Wang, O., Fenty, I., Forget, G., Heimbach, P., and Ponte, R.: Synopsis of the ECCO central production global Ocean and sea-ice state estimate, version 4 release 4, https://doi.org/10.5281/zenodo.3765929, retrieved from, https://ecco.jpl.nasa.gov/drive/files/Version4/Release4, 2021.

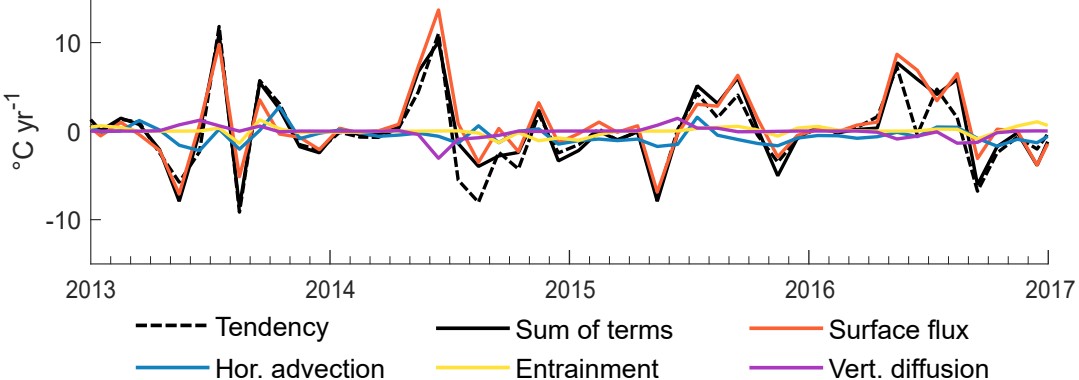

**Figure A6.** Anomalies in the dominant terms of the mixed layer temperature budget, relative to the 1992-2015 monthly climatology (°C yr$^{-1}$) for ECCOv4-r4, averaged over the northeast of the North Atlantic (35-5°W, 53-63°N). Anomalies in the model temperature tendency are shown by the dashed black line, and anomalies in the sum of the temperature budget terms driving the change in temperature by the solid black line. The remaining lines represent those individual processes: the surface heat flux (orange), horizontal advection (blue), vertical entrainment (yellow), and vertical diffusion (purple).

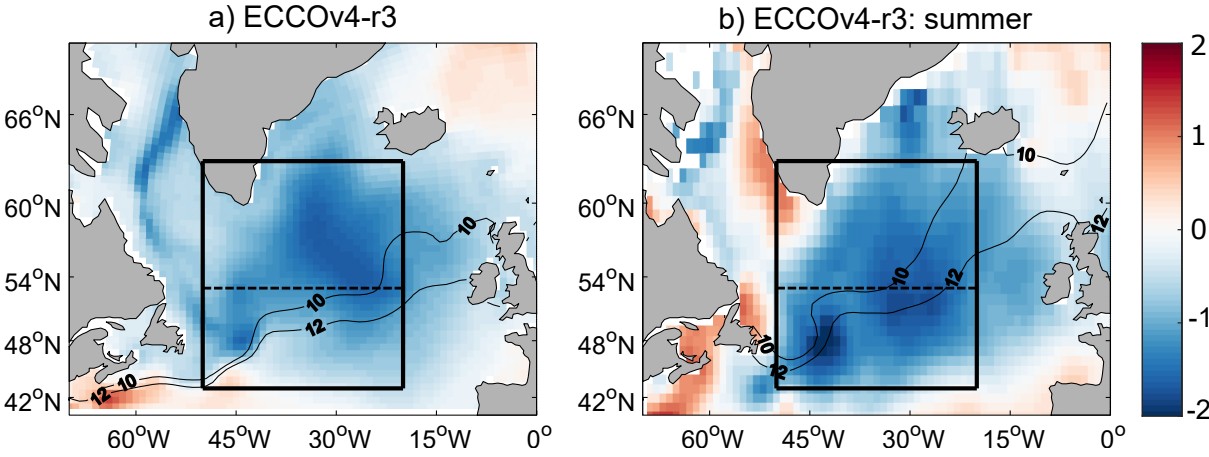

**Figure A7.** The SST anomaly (°C) in the ECCOv4-r3 monthly SST observations, relative to the 1992-2015 climatology, averaged over a) the whole of 2015 and b) the summer (JJA) only. The black boxes mark the region we use to define the extent of the 2015 cold anomaly (50-20°W, 43-63°N). The average position of the 10°C and 12°C isotherms for each period is also shown.

Forget, G., Campin, J.-M., Heimbach, P., Hill, C. N., Ponte, R. M., and Wunsch, C.: ECCO version 4: An integrated framework for non-linear inverse modeling and global ocean state estimation, Geoscientific Model Development, 8, 3071–3104, https://doi.org/10.5194/gmd-8-3071-2015, 2015a.

Forget, G., Ferreira, D., and Liang, X.: On the observability of turbulent transport rates by Argo: supporting evidence from an inversion experiment, Ocean Science, 11, 839–853, https://doi.org/10.5194/os-11-839-2015, 2015b.

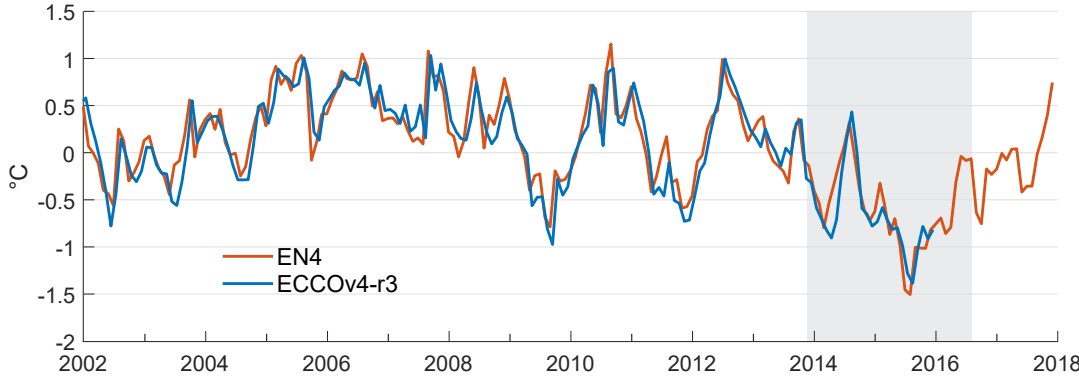

**Figure A8.** Time series of anomalies in the mixed layer average potential temperature (°C), relative to the 1992-2015 monthly climatology, averaged over the cold blob region, for EN4 observations (red) and ECCOv4-r3 (blue). The shaded area marks the time period from when the initial cold anomaly begins to emerge to when the anomaly once again becomes positive.

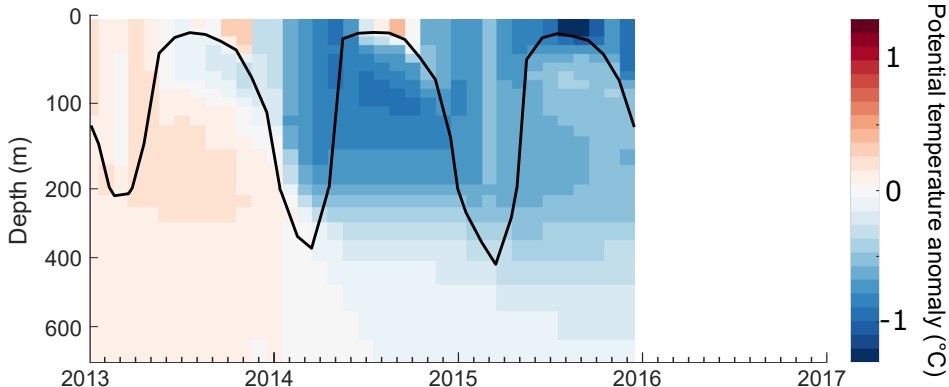

**Figure A9.** The potential temperature anomaly over depth relative to the 1992-2015 monthly climatology (color; °C), averaged over the cold blob region in ECCOv4-r3. The MLD is also shown (black line, m). Note the non-uniform spacing of the vertical axes.

Foukal, N. P. and Lozier, M. S.: Examining the origins of ocean heat content variability in the eastern North Atlantic subpolar gyre, Geophysical Research Letters, 45, 11–275, https://doi.org/10.1029/2018GL079122, 2018.

Frankignoul, C.: Sea surface temperature anomalies, planetary waves, and air-sea feedback in the middle latitudes, Reviews of geophysics, 23, 357–390, https://doi.org/10.1029/RG023i004p00357, 1985.

Fukumori, I., Wang, O., Fenty, I., Forget, G., Heimbach, P., and Ponte, R. M.: ECCO version 4 release 3, Tech. rep., https://dspace.mit.edu/handle/1721.1/110380, 2017.

Gaspar, P. and Grégoris, Y.: A simple eddy-kinetic-energy model for simulations of the ocean vertical mixing: tests at station Papa and

long-term upper ocean study site, Journal of Geophysical Research, 95, 16–179, https://doi.org/10.1029/JC095iC09p16179, 1990.

Gent, P. R. and McWilliams, J. C.: Isopycnal mixing in ocean circulation models, Journal of Physical Oceanography, 20, 150–155, https://doi.org/10.1175/1520-0485(1990)020<0150:IMIOCM>2.0.CO;2, 1990.

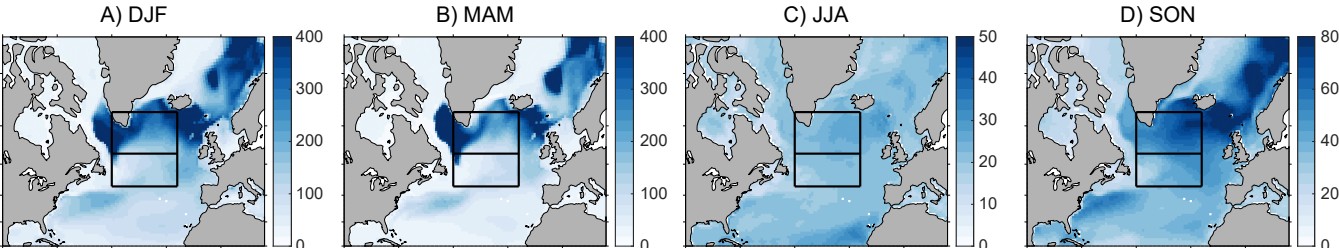

**Figure A10.** The average seasonal cycle of North Atlantic MLD (m) from 1992-2015 in ECCOv4-r3. Note the varying color scales between seasons.

Grist, J. P. and Josey, S. A.: Inverse analysis adjustment of the SOC air–sea flux climatology using ocean heat transport constraints, Journal of Climate, 16, 3274–3295, https://doi.org/https://doi.org/10.1175/1520-0442(2003)016<3274:IAAOTS>2.0.CO;2, 2003.

Grist, J. P., Josey, S. A., Jacobs, Z. L., Marsh, R., Sinha, B., and Van Sebille, E.: Extreme air–sea interaction over the North Atlantic subpolar gyre during the winter of 2013–2014 and its sub-surface legacy, Climate dynamics, 46, 4027–4045, https://doi.org/10.1007/s00382-015-2819-3, 2016.

Hátún, H., Olsen, B., and Pacariz, S.: The dynamics of the North Atlantic subpolar gyre introduces predictability to the breeding success of kittiwakes, Frontiers in Marine Science, 4, 123, https://doi.org/10.3389/fmars.2017.00123, 2017.

Holliday, N. P., Bersch, M., Berx, B., Chafik, L., Cunningham, S., Florindo-López, C., Hátún, H., Johns, W., Josey, S. A., Larsen, K. M. H., et al.: Ocean circulation causes the largest freshening event for 120 years in eastern subpolar North Atlantic, Nature Communications, 11, 1–15, https://doi.org/10.1038/s41467-020-14474-y, 2020.

Josey, S., De Jong, M., Oltmanns, M., Moore, G., and Weller, R.: Extreme variability in Irminger Sea winter heat loss revealed by ocean observatories initiative mooring and the ERA5 reanalysis, Geophysical Research Letters, 46, 293–302, https://doi.org/10.1029/2018GL080956, 540    2019.

Josey, S. A., Hirschi, J. J.-M., Sinha, B., Duchez, A., Grist, J. P., and Marsh, R.: The recent Atlantic cold anomaly: Causes, consequences, and related phenomena, Annual review of marine science, 10, 475–501, https://doi.org/10.1146/annurev-marine-121916-063102, 2018.

Kostov, Y., Johnson, H. L., Marshall, D. P., Heimbach, P., Forget, G., Holliday, N. P., Lozier, M. S., Li, F., Pillar, H. R., and Smith, T.: Distinct sources of interannual subtropical and subpolar Atlantic overturning variability, Nature Geoscience, pp. 1–5, 545    https://doi.org/10.1038/s41561-021-00759-4, 2021.

Lamb, P. J. and Peppler, R. A.: North Atlantic Oscillation: concept and an application, Bulletin of the American Meteorological Society, 68, 1218–1225, https://doi.org/10.1175/1520-0477(1987)068<1218:NAOCAA>2.0.CO;2, 1987.

Large, W. and Yeager, S.: The global climatology of an interannually varying air–sea flux data set, Climate dynamics, 33, 341–364, https://doi.org/10.1007/s00382-008-0441-3, 2009.

Lindsay, R., Wensnahan, M., Schweiger, A., and Zhang, J.: Evaluation of seven different atmospheric reanalysis products in the Arctic, Journal of Climate, 27, 2588–2606, https://doi.org/10.1175/JCLI-D-13-00014.1, 2014.

Losch, M., Menemenlis, D., Campin, J.-M., Heimbach, P., and Hill, C.: On the formulation of sea-ice models. Part 1: Effects of different solver implementations and parameterizations, Ocean Modelling, 33, 129–144, https://doi.org/10.1016/j.ocemod.2009.12.008, 2010.

Maroon, E. A., Yeager, S. G., Danabasoglu, G., and Rosenbloom, N.: Was the 2015 North Atlantic subpolar cold anomaly predictable?, 555    Journal of Climate, pp. 1–69, https://doi.org/10.1175/JCLI-D-20-0750.1, 2021.

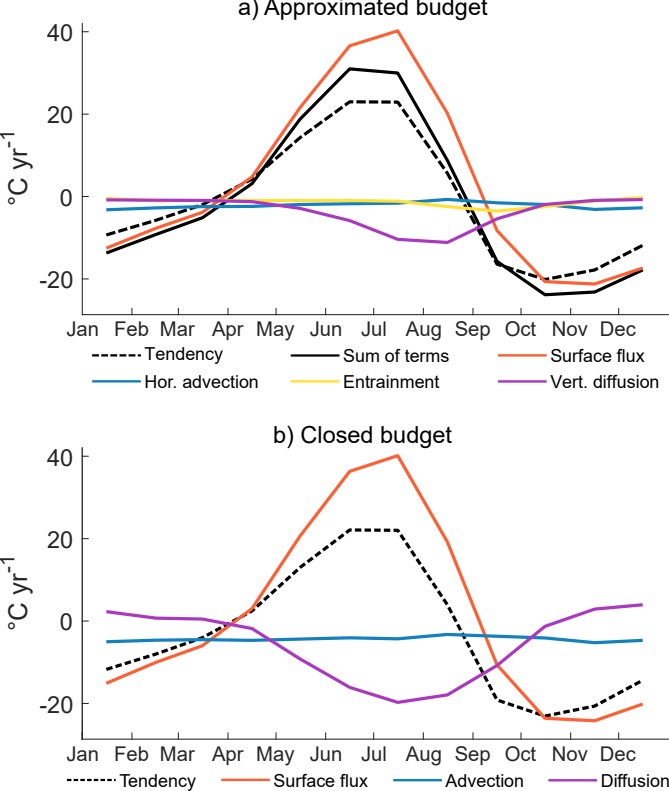

**Figure A11.** The average seasonal cycle of the dominant mixed layer temperature budget terms (°C yr$^{-1}$) averaged over the cold blob region for ECCOv4-r3, where positive values represent an increase in the rate of change in the temperature of the mixed layer. a) The approximated budget computed using Equation 1, where the black dashed line is the actual temperature tendency in the model, and the solid line is the sum of the budget terms driving that temperature change. The remaining lines represent the temperature change due to each individual process: surface heat fluxes (orange), horizontal advection (blue), vertical entrainment (yellow), and vertical diffusion (purple). Horizontal diffusion and lateral induction are not shown as the effects of both are negligible. b) The closed budget computed online for comparison, where the dashed line is the tendency, which is equal to the sum of surface fluxes (red), advection (blue), and diffusion (purple).

Marshall, J., Johnson, H., and Goodman, J.: A study of the interaction of the North Atlantic Oscillation with ocean circulation, Journal of Climate, 14, 1399–1421, https://doi.org/10.1175/1520-0442(2001)014<1399:ASOTIO>2.0.CO;2, 2001.

Mecking, J., Drijfhout, S., Hirschi, J. J., and Blaker, A.: Ocean and atmosphere influence on the 2015 European heatwave, Environmental Research Letters, 14, 114 035, https://doi.org/10.1088/1748-9326/ab4d33, 2019.

Moore, G.: Gale force winds over the Irminger Sea to the east of Cape Farewell, Greenland, Geophysical Research Letters, 30, https://doi.org/10.1029/2003GL018012, 2003.

Peter, A.-C., Le Hénaff, M., Du Penhoat, Y., Menkes, C. E., Marin, F., Vialard, J., Caniaux, G., and Lazar, A.: A model study of the seasonal mixed layer heat budget in the equatorial Atlantic, Journal of Geophysical Research: Oceans, 111, https://doi.org/10.1029/2005JC003157, 2006.

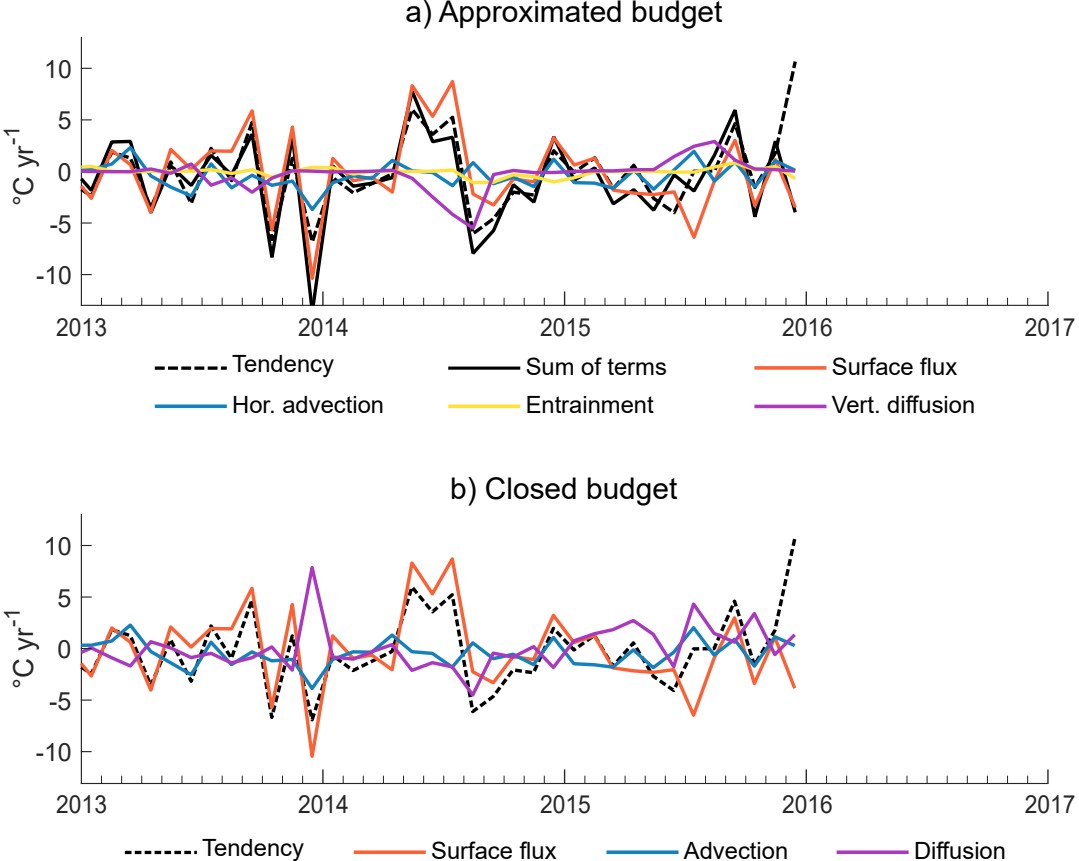

**Figure A12.** Anomalies in the dominant terms of the mixed layer temperature budget, relative to the 1992-2015 monthly climatology (°C yr$^{-1}$), averaged over the cold blob region. a) Anomalies in the approximated budget, where the dashed black line shows anomalies in the model temperature tendency, and the solid black line shows anomalies in the sum of the temperature budget terms driving the temperature change. The remaining lines represent those individual processes: the surface heat flux (orange), horizontal advection (blue), vertical entrainment (yellow), and vertical diffusion (purple). Lateral induction and horizontal diffusion are not shown because anomalies in both are negligible. b) Anomalies in the closed budget.

Piecuch, C. G.: A note on practical evaluation of budgets in ECCO version 4 release 3, Tech. rep., http://hdl.handle.net/1721.1/111094, 2017.

Piecuch, C. G., Ponte, R. M., Little, C. M., Buckley, M. W., and Fukumori, I.: Mechanisms underlying recent decadal changes in subpolar North Atlantic Ocean heat content, Journal of Geophysical Research: Oceans, 122, 7181–7197, https://doi.org/10.1002/2017JC012845, 2017.

Piron, A., Thierry, V., Mercier, H., and Caniaux, G.: Gyre-scale deep convection in the subpolar North Atlantic Ocean during winter 2014–

2015, Geophysical Research Letters, 44, 1439–1447, https://doi.org/10.1002/2016GL071895, 2017.

Rahmstorf, S., Box, J. E., Feulner, G., Mann, M. E., Robinson, A., Rutherford, S., and Schaffernicht, E. J.: Exceptional twentieth-century slowdown in Atlantic Ocean overturning circulation, Nature climate change, 5, 475–480, https://doi.org/10.1038/nclimate2554, 2015.

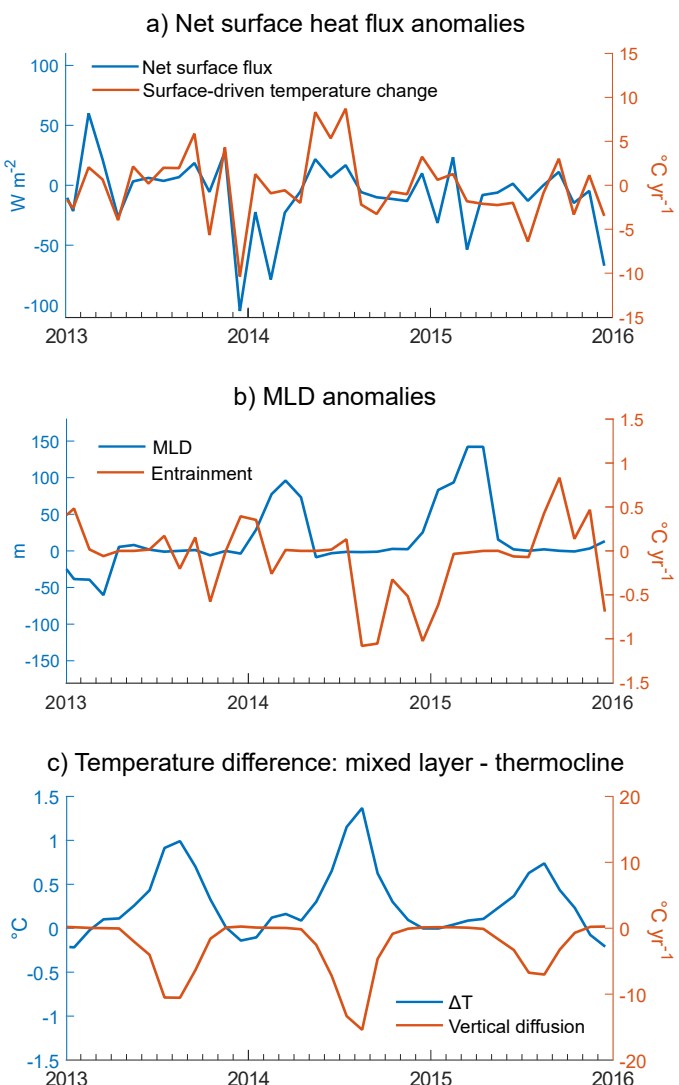

**Figure A13.** Time series of the individual components that comprise the dominant temperature budget terms, and the associated change in mixed layer temperature, both averaged over the cold blob region in ECCOv4-r3. a) Anomalies in the heat flux into the surface output by the model, defined as $Q_{net}$ in Equation 1 (W m$^{-2}$; blue), and anomalies in the associated change in mixed layer temperature, i.e. the surface flux term of the mixed layer budget (°C yr$^{-1}$; red). b) Anomalies in MLD (m; blue) and the associated heat entrainment term of the mixed layer budget (°C yr$^{-1}$; red). c) The temperature difference between the mixed layer and the model cell immediately beneath (°C; blue), defined as $\Delta T$ in Equation 1 where positive values signify that the mixed layer is warmer than the thermocline, and the associated vertical diffusion term of the mixed layer budget (°C yr$^{-1}$; red). Note, the seasonal cycle has not been removed from either term in c).

Rayner, N., Parker, D. E., Horton, E., Folland, C. K., Alexander, L. V., Rowell, D., Kent, E., and Kaplan, A.: Global analyses of sea surface temperature, sea ice, and night marine air temperature since the late nineteenth century, Journal of Geophysical Research: Atmospheres, 575      108, https://doi.org/10.1029/2002JD002670, 2003.

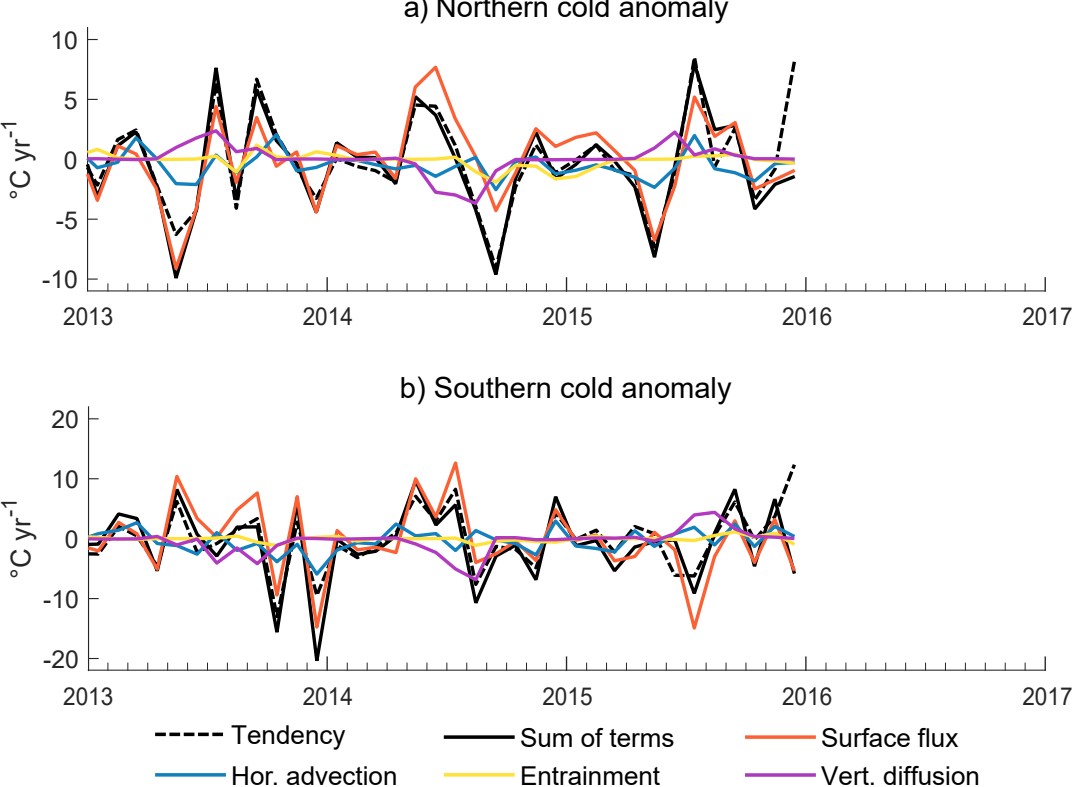

**Figure A14.** Anomalies in the dominant terms of the mixed layer temperature budget, relative to the 1992-2015 monthly climatology (°C yr$^{-1}$) for ECCOv4-r3. The anomalies in the model temperature tendency are shown by the dashed black line, and the anomalies in the sum of the temperature budget terms driving the change in temperature by the solid black line. The remaining lines represent those individual terms: the surface heat flux (orange), horizontal advection (blue), vertical entrainment (yellow), and vertical diffusion (purple). The results are averaged over a) the northern half of the cold blob region (50-20°W, 53-63°N) and b) the southern half of the cold anomaly (50-20°W, 43-53°N).

Rogers, J. C.: The association between the North Atlantic Oscillation and the Southern Oscillation in the northern hemisphere, Monthly Weather Review, 112, 1999–2015, https://doi.org/10.1175/1520-0493(1984)112<1999:TABTNA>2.0.CO;2, 1984.

Sutton, R. and Mathieu, P.-P.: Response of the atmosphere–ocean mixed-layer system to anomalous ocean heat-flux convergence, Quarterly Journal of the Royal Meteorological Society: A journal of the atmospheric sciences, applied meteorology and physical oceanography, 128, 1259–1275, https://doi.org/10.1256/003590002320373283, 2002.

Taws, S. L., Marsh, R., Wells, N. C., and Hirschi, J.: Re-emerging ocean temperature anomalies in late-2010 associated with a repeat negative NAO, Geophysical Research Letters, 38, https://doi.org/10.1029/2011GL048978, 2011.

Tesdal, J.-E., Abernathey, R. P., Goes, J. I., Gordon, A. L., and Haine, T. W.: Salinity trends within the upper layers of the subpolar North Atlantic, Journal of Climate, 31, 2675–2698, https://doi.org/10.1175/JCLI-D-17-0532.1, 2018.

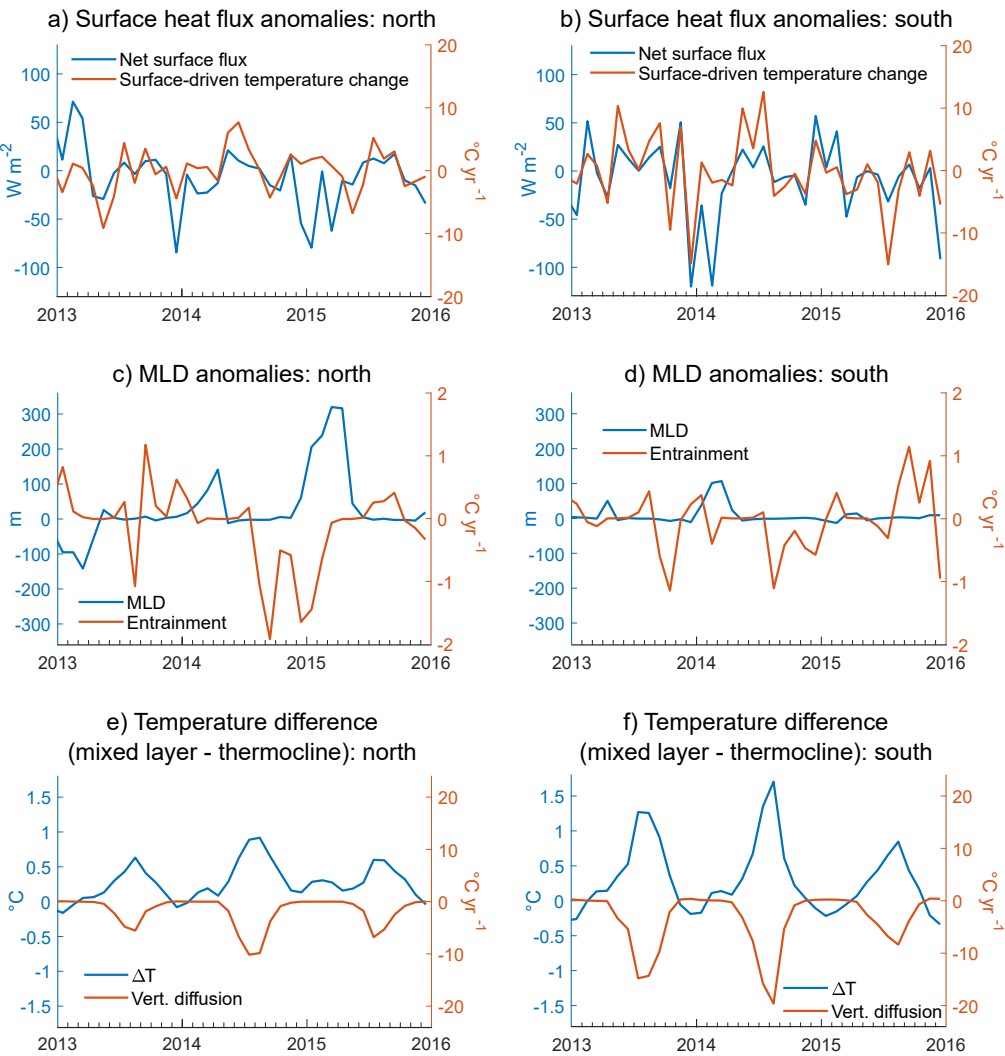

**Figure A15.** Time series of the individual components that comprise the dominant temperature budget terms, and the associated change in mixed layer temperature, both averaged over the north (left) and south (right) of the cold blob region in ECCOv4-r3. a) Anomalies in the heat flux into the surface output by the model, defined as $Q_{net}$ in Equation 1 (W m$^{-2}$; blue), and anomalies in the associated change in mixed layer temperature, i.e. the surface flux term of the mixed layer budget (°C yr$^{-1}$; red). b) Anomalies in MLD (m; blue) and the associated heat entrainment term of the mixed layer budget (°C yr$^{-1}$; red). c) The temperature difference between the mixed layer and the model cell immediately beneath (°C; blue), defined as $\Delta T$ in Equation 1 where positive values signify that the mixed layer is warmer than the thermocline, and the associated vertical diffusion term of the mixed layer budget (°C yr$^{-1}$; red). Note, the seasonal cycle has not been removed from either term in e-f).

Wallace, J. M. and Gutzler, D. S.: Teleconnections in the geopotential height field during the Northern Hemisphere winter, Monthly weather review, 109, 784–812, https://doi.org/10.1175/1520-0493(1981)109<0784:TITGHF>2.0.CO;2, 1981.

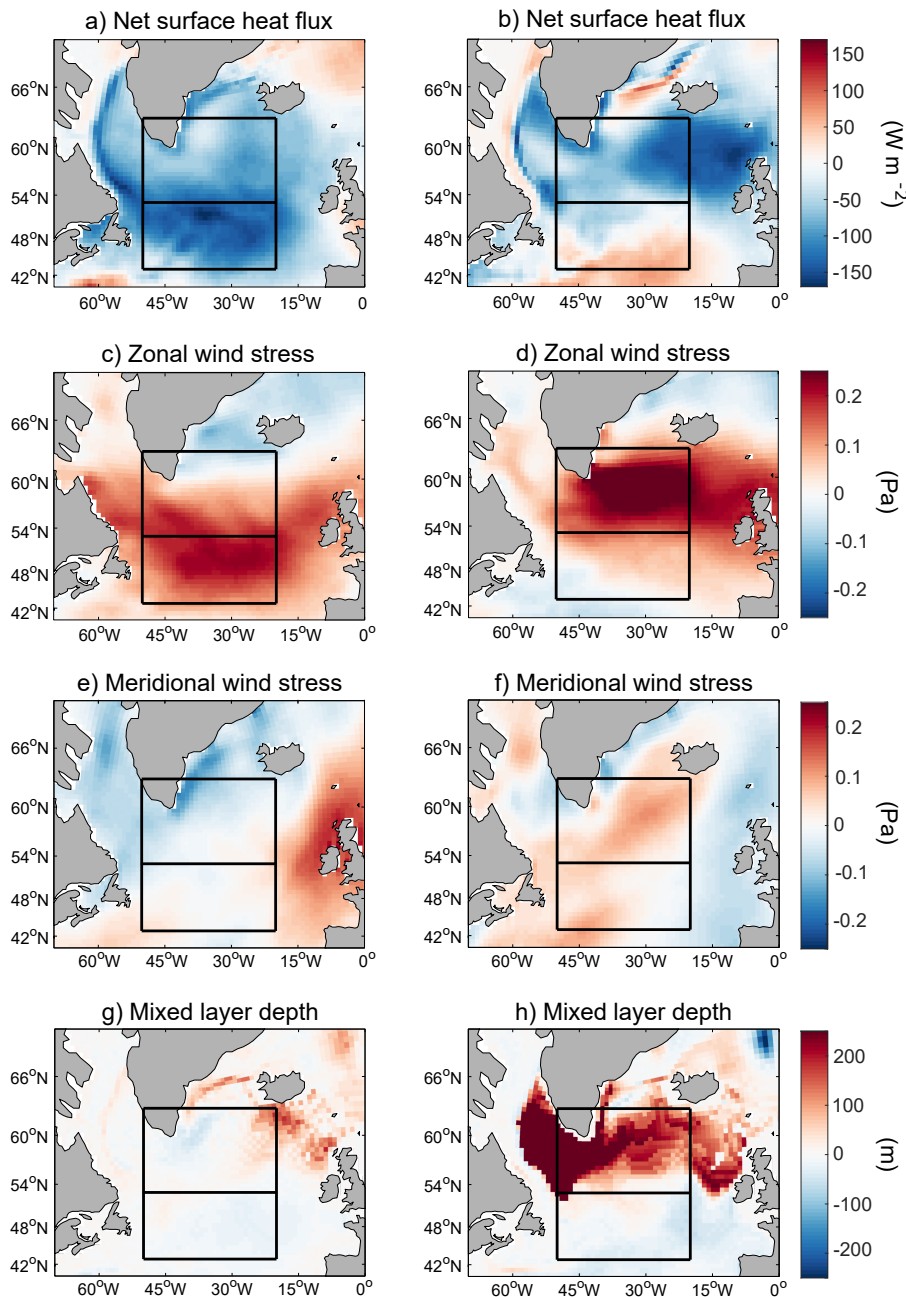

**Figure A16.** The spatial distribution of ECCOv4-r3 anomalies in the terms causing the initial anomalous surface cooling in December 2013 (left) and the same anomalies for January 2015 (right), when the net heat flux out of the surface ocean was also high but its impact not seen in the temperature of the mixed layer. Shown are anomalies in a,b) the net surface heat flux (W m$^{-2}$), c,d) zonal wind stress (Pa), e,f) meridional wind stress (Pa), and g,h) MLD (m).

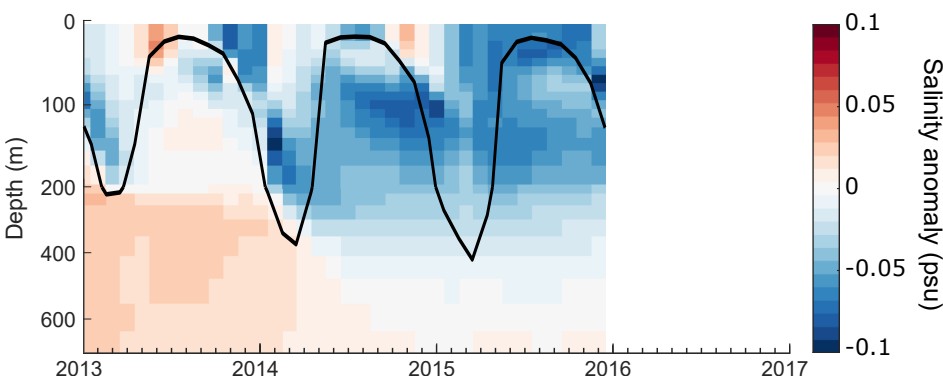

**Figure A17.** Salinity anomalies over depth relative to the 1992-2015 monthly climatology (color; psu), averaged over the cold blob region in ECCOv4-r3 for the period leading up to and during the 2015 cold anomaly. The MLD is also shown (black line, m). Note the non-uniform spacing of the vertical axis.

Yeager, S. G., Kim, W. M., and Robson, J.: What caused the Atlantic cold blob of 2015?, US CLIVAR Variations, 14, 24–31, 2016.