# Peer review of "Causes of the 2015 North Atlantic cold anomaly in the ECCOv4 state estimate"

_Ocean Science, 2022_

## Referee Comment (RC1)

Review of "Causes of the 2015 North Atlantic cold anomaly in the ECCOv4 state estimates" by Sanders et al.

The manuscript presents an in-depth analysis of a large and cold SST anomaly in 2015 in the eastern North Atlantic subpolar gyre. This signal has been identified previously, and this paper adds to that literature by decomposing the mixed layer heat budget during the cold anomaly. The authors conclude that the majority of the signal is attributable to surface heat fluxes, though horizontal advection and re-emergence from temperature signals below the mixed layer also played a significant role.

I found the paper to be an important addition to the literature. The text was well-written, the logic was clear, and the figures were appropriate. It was clear the authors have put a lot of thought and effort into the submission, which is much appreciated. I have listed a number of important topics the authors should address prior to publication, but none rise to the level of major revisions. I therefore recommend the paper be published pending minor revisions.

1. The authors should clarify in the introduction how this study adds to the results from Grist et al. (2016) and Josey et al. (2018). What is new about these findings compared to those papers?

2. It is still unclear to me what the rationale is for using ECCOv4r3. If there aren't any significant differences then why not use the more recent release? Using a different data product (e.g., EN4) would provide some perspective on the range of values to expect from different reanalyses/state estimates, but using an outdated and updated version of the same product doesn't provide any new information. It also appears the authors abandon r3 half-way through the manuscript (with those figures included in the appendix), so either the authors need to make this case stronger for r3 (probably around line 90) or only conduct the analysis in r4 (my preference).

3. I thought the main utility of ECCO was that it conserved energy/heat! Why don't the seasonal budgets close? It appears the authors try to explain this in the first paragraph of Section 4.5, but this is too late in the manuscript. I recommend this explanation should be moved to the introduction, potentially with more information included because why the heat budgets did not close was still not clear to me… Doesn't the heat budget for each grid cell close? If so, how can the heat budget over multiple grid cells not close?

4. I recommend combining figs 1 and 2 to make comparisons to the observations easier. I also recommend acknowledging a pretty apparent 'NAC' shape in the cold anomalies in Fig. 2c and 2d that is not present in the r3 (Fig. 2a and 2b) or observations (Fig. 1). That suggests that the NAC has shifted southward in r4... is that a correct interpretation? I also think it would be useful to include figures of ECCO SST-HadISST to see how close ECCO SST is to the observations. Similarly, this could be done for MLDs in ECCO and observations.

5. Fig. 3 – Can the mixed layer temperature from EN4 be added to Fig. 3?

6. Fig. 4 – I am a bit surprised that there is a clear signal of re-emergence when you're averaging the MLD over such a large and inhomogeneous region. This is why it would be nice to see a map of MLD in ECCO versus the observations to compare whether ECCO is getting the MLD correct.

7. Line 181 – Can you elaborate on how the cold anomaly drives increased convection? Through preconditioning the stratification below the mixed layer or are you suggesting the fluxes are altered by the ocean temperature anomaly?

8. Fig. 6 (and other time series) – it is hard to see the individual curves over course of 4 years… can you zoom in on the 2014-2016 period?

9. Fig. 7 – What is the difference (if any) between Fig. 7a and Fig. 6a?

10. Fig. 7 – I was also a bit confused by Fig. 7c – it took me a while to understand that the 'bump' in the blue line (vertical temperature gradient at the base of the mixed layer) in winter 2014 is an important signal. Can you do something to highlight that? Otherwise, the most obvious signal for the reader to notice is the anticorrelation between the red and blue curves, but that result is pretty obvious considering the vertical diffusion out of the mixed layer should be proportional to the vertical temperature gradient at the base of the mixed layer.

11. Fig. 8 – Nice schematic! Very clear. Two suggestions: (1) replace "Mixed layer shallows" to "Mixed layer shoals due to seasonal cycle". At first, I wasn't sure if you were still talking about anomalies from the seasonal cycle, so I was confused why the mixed layer would shoal in the spring. And (2) flip the red/blue gradient in the first panel – do you mean to imply cold water will overly warm water or just that the surface waters are cooling relative to the waters near the base of the mixed layer?

12. Fig. 9 – Could you mark the 53°N line that separates the Northern/Southern regions on Fig. 1 and/or Fig. 2?

13. Lines 269-273. While it is interesting that advection plays a larger role in the south, I'm not sure I believe ECCO is best suited to make these comparisons. I would tend to believe the ECCO heat budgets in the northern region more than southern region due to issues resolving the GS/NAC system. In particular, I'm thinking that those 'NAC' shaped anomalies in Fig. 2 that are not apparent in the observations may be playing a disproportionate role in ECCOv4r4 than is present in the real ocean.

14. Fig. 11 – Another really nice plot/schematic, but I was a bit confused by the colors in the background. I thought they corresponded to the mechanisms in the bottom legend. I think they're actually just showing the seasons (is that correct?). I suggest either using a different color scheme or just dashed lines to demarcate the different seasons.

15. Section 4.5 – I suggested above that the first paragraph should go in the introduction, and I think the second paragraph should have its own heading along the lines of "Caveats of ECCO in this region". In addition to this second paragraph, there should also be a list of potential biases in ECCO for this region. The GS/NAC position (especially in the NW Corner region) is the first one that comes to mind, but also the overflows at Greenland Scotland Ridge is another important one. Another one would be that the shelf circulation is not resolved well in ECCO. ECCO is a great tool but there are caveats that go along with working with it, and the text will read better if you acknowledge those caveats up front rather than let the reader ponder their effects on their own.

16. I found that the text overlooked a relatively well-developed literature that has used ECCOv4 in this region: Piecuch et al. (2017), Foukal and Lozier (2018), Tesdal et al. (2019), and Asbjornsen et al. (2020). Including these papers could strengthen the authors case that ECCO is well-suited for this region.

Minor edits:
Lines 14-24 – It was a bit strange to repeat the last sentence in the abstract as the second sentence of the introduction. Can you just remove the sentence from the Introduction?

Line 77 – what is 1 x 1/3-1? I know ECCO's grid size varies with respect to latitude, but this notation is not clear.

Line 195 (and throughout) – It's a bit confusing to cite the numbers in the text as °C/month when they're listed as °C/year in the figure (and sometimes in the text). Can you just use one unit throughout?

Fig. 10f – please remove one of the two legends in this panel (but I'll admit two legends are better than no legends!).

Nick Foukal
Woods Hole Oceanographic Institution

References:
Piecuch, C. G., Ponte, R. M., Little, C. M., Buckley, M. W., and Fukumori, I. (2017). Mechanisms underlying recent decadal changes in subpolar North Atlantic ocean heat content. Journal of Geophysical Research, doi: 10.1002/2017JC012845.
Foukal, N. P., and Lozier, M. S. (2018). Examining the origins of ocean heat content variability in the eastern North Atlantic subpolar gyre. Geophysical Research Letters, doi: 10.1029/2018GL079122.
Tesdal, J. E., Abernathey, R. P., Goes, J. I., Gordon, A. L., and Haine, T. (2018). Salinity trends within the upper layers of the subpolar North Atlantic, Journal of Climate, doi: 10.1175/JCLI-D-17-0532.1
Asbjornsen, H., Arthun, M., Skagseth, O., and Eldevik, T. (2019) Mechanisms of ocean heat anomalies in the Norwegian Sea, Journal of Geophysical Research, doi: 10.1029/2018JC014649.

---

## Author Comment (AC1)

**Response to Reviewer 1**

Thank you for your helpful and constructive comments on our paper. The comments are listed below in black, with our response to each in blue. We hope that you are happy with the improved manuscript and find it suitable for publication.

The manuscript presents an in-depth analysis of a large and cold SST anomaly in 2015 in the eastern North Atlantic subpolar gyre. This signal has been identified previously, and this paper adds to that literature by decomposing the mixed layer heat budget during the cold anomaly. The authors conclude that the majority of the signal is attributable to surface heat fluxes, though horizontal advection and re-emergence from temperature signals below the mixed layer also played a significant role.

I found the paper to be an important addition to the literature. The text was well-written, the logic was clear, and the figures were appropriate. It was clear the authors have put a lot of thought and effort into the submission, which is much appreciated. I have listed a number of important topics the authors should address prior to publication, but none rise to the level of major revisions. I therefore recommend the paper be published pending minor revisions.

Thank you for the encouraging comments, we have endeavoured to improve the paper based on your recommendations.

1. The authors should clarify in the introduction how this study adds to the results from Grist et al. (2016) and Josey et al. (2018). What is new about these findings compared to those papers?

This has now been clarified in the final paragraph of the introduction (lines 55-61).

2. It is still unclear to me what the rationale is for using ECCOv4r3. If there aren't any significant differences then why not use the more recent release? Using a different data product (e.g., EN4) would provide some perspective on the range of values to expect from different reanalyses/state estimates, but using an outdated and updated version of the same product doesn't provide any new information. It also appears the authors abandon r3 half-way through the manuscript (with those figures included in the appendix), so either the authors need to make this case stronger for r3 (probably around line 90) or only conduct the analysis in r4 (my preference).

We agree that the inclusion of ECCOv4-r3 was not well-justified. The analysis of ECCOv4-r3 has therefore been removed from the main text, and figures showing the results in release 3 have instead been included in the Appendix.

3. I thought the main utility of ECCO was that it conserved energy/heat! Why don't the seasonal budgets close? It appears the authors try to explain this in the first paragraph of Section 4.5, but this is too late in the manuscript. I recommend this explanation should be moved to the introduction, potentially with more information included because why the heat

budgets did not close was still not clear to me… Doesn't the heat budget for each grid cell close? If so, how can the heat budget over multiple grid cells not close?

We chose to use a well-established analysis method that allows for a separation between entrainment and vertical diffusion following the movement of the mixed layer; this method allows us to identify the relative importance of these processes, leading to one of the key points of this paper regarding the dominance of vertical diffusion in the re-emergence process. Because of how this method is constructed, the budgets do not close perfectly either for individual grid cells or for the mixed layer. This is primarily because the of the assumptions and parameterisations involved in the budget equation, as well as the resolution of the data used to compute the budgets. We agree that this explanation should be moved into the introduction. Section 4.5 discussing the computation of mixed layer budgets in ECCO has now been moved to the introduction, and more detail added (lines 61-75). The closed budgets have now also been included within the main text for comparison with the approximated budgets (Fig. 5b and 6b).

4. I recommend combining figs 1 and 2 to make comparisons to the observations easier. I also recommend acknowledging a pretty apparent 'NAC' shape in the cold anomalies in Fig. 2c and 2d that is not present in the r3 (Fig. 2a and 2b) or observations (Fig. 1). That suggests that the NAC has shifted southward in r4... is that a correct interpretation? I also think it would be useful to include figures of ECCO SSTHadISST to see how close ECCO SST is to the observations. Similarly, this could be done for MLDs in ECCO and observations.

Fig. 1 and 2 have now been combined. The time series of ECCO SST and that of HadISST observations are shown in a schematic in Fig. 11 – we chose not to include them earlier in the paper because the time series is very similar to that of mixed layer average temperature already shown in Fig. 2. A time series of MLD in the EN4 observations has now been added for comparison with the model (Fig. 3).

Since we are no longer including the results from ECCOv4-r3 in the text, we have not gone into more detail for the potential differences between the models. However, we have plotted the surface currents in both and found them to be almost identical, suggesting any differences in the SST patterns are not due to a shift in the NAC between the two models. The 10° and 12° isotherms have also been added to the SST maps in Fig. 1, with strong similarities between ECCOv4-r4 and observations, also suggesting the position of the surface currents in ECCOv4 is close to that of observations.

5. Fig. 3 – Can the mixed layer temperature from EN4 be added to Fig. 3?

Thank you for the suggestion, the mixed layer temperature and MLD calculated in EN4 has now been added to Fig 2 (previously Fig 3).

6. Fig. 4 – I am a bit surprised that there is a clear signal of re-emergence when you're averaging the MLD over such a large and inhomogeneous region. This is why it would be nice to see a map of MLD in ECCO versus the observations to compare whether ECCO is getting the MLD correct.

Maps showing the average seasonal cycle in MLD computed in EN4 and ECCOv4-r4 have now been added (Fig. 4). The time series of MLD averaged over the cold anomaly region is also shown in Fig. 3 for EN4 to compare with that of ECCOv4-r4. Overall, there is strong similarity between the mixed layers in the model and that in observations. This has been discussed in lines 201-206.

7. Line 181 – Can you elaborate on how the cold anomaly drives increased convection? Through preconditioning the stratification below the mixed layer or are you suggesting the fluxes are altered by the ocean temperature anomaly?

This has now been added (lines 199-201).

8. Fig. 6 (and other time series) – it is hard to see the individual curves over course of 4 years… can you zoom in on the 2014-2016 period?

We prefer not to remove the two years from this specific plot as the anomalies start partway through 2013 and the processes causing the diminishment of the anomaly occur through 2016. However, we appreciate that the budget time series are quite cluttered and hope that the following changes make the figure easier to read: the legend has been moved to below the figure, the grid lines removed, the spacing of the dashed line altered to be clearer, and the colours of the lines changed. The time series showing the budget components (Fig. 7, Fig. 10) have now been edited to only show 2013-2016, but we still include 2013 because the surface forcing at the end of 2013 is particularly important.

9. Fig. 7 – What is the difference (if any) between Fig. 7a and Fig. 6a?

The blue line in Fig. 7a is the same as that showing anomalies in the surface flux term of the mixed layer budget in Fig. 6. This term is dependent on both the net heat flux through the surface (red line in Fig. 7a) and MLD (blue line in Fig. 7b). The term is repeated in this figure in order to show that there is not perfect correlation between the anomalies in the net heat fluxes and the surface-driven temperature change, due to the impact of the MLD on the volume of water being heated/cooled. This has now been made clearer in the text (lines 240-244).

10. Fig. 7 – I was also a bit confused by Fig. 7c – it took me a while to understand that the 'bump' in the blue line (vertical temperature gradient at the base of the mixed layer) in winter 2014 is an important signal. Can you do something to highlight that? Otherwise, the most obvious signal for the reader to notice is the anticorrelation between the red and blue curves, but that result is pretty obvious considering the vertical diffusion out of the mixed layer should be proportional to the vertical temperature gradient at the base of the mixed layer.

The aim of Fig. 7c is to show that during the cold anomaly, the temperature difference across the base of the mixed layer is particularly high (due to the cold anomaly being sequestered below the mixed layer and surface warming of the water above), leading to the strong diffusive cooling. The increase in the $\Delta T$ term during winter 2014, due to the sequestration of the cold anomaly and further cooling below the mixed layer, while interesting does not lead

to cooling via diffusion. This is because the mixed layer was anomalously deep during this time, masking the impact of the increased $\Delta T$. This has now been discussed in lines 253-255.

11. Fig. 8 – Nice schematic! Very clear. Two suggestions: (1) replace "Mixed layer shallows" to "Mixed layer shoals due to seasonal cycle". At first, I wasn't sure if you were still talking about anomalies from the seasonal cycle, so I was confused why the mixed layer would shoal in the spring. And (2) flip the red/blue gradient in the first panel – do you mean to imply cold water will overly warm water or just that the surface waters are cooling relative to the waters near the base of the mixed layer?

Thank you for the suggestions, both changes have now been made. Initially, the blue over red was meant to show that the surface waters are cooling relative to the deeper water below the mixed layer, but we agree that was confusing.

12. Fig. 9 – Could you mark the 53°N line that separates the Northern/Southern regions on Fig. 1 and/or Fig. 2?

The 53°N line has been added to each map throughout the paper.

13. Lines 269-273. While it is interesting that advection plays a larger role in the south, I'm not sure I believe ECCO is best suited to make these comparisons. I would tend to believe the ECCO heat budgets in the northern region more than southern region due to issues resolving the GS/NAC system. In particular, I'm thinking that those 'NAC' shaped anomalies in Fig. 2 that are not apparent in the observations may be playing a disproportionate role in ECCOv4r4 than is present in the real ocean.

The position of the 10°C and 12°C isotherms within the North Atlantic are very similar between ECCO and EN4 (Fig. 1), suggesting that the position of the NAC in ECCOv4-r4 is actually close to observations. It is therefore unlikely that the position of the NAC in the model is causing the impact of advection in the region to be disproportionately wrong.

14. Fig. 11 – Another really nice plot/schematic, but I was a bit confused by the colors in the background. I thought they corresponded to the mechanisms in the bottom legend. I think they're actually just showing the seasons (is that correct?). I suggest either using a different color scheme or just dashed lines to demarcate the different seasons.

Thank you for the suggestion. The colours of the arrows have been changed to match the new colours of the lines representing the same processes in the budget figures. The background colours have also been made paler and the months labelled at the top in the same colours, to hopefully avoid any confusion.

15. Section 4.5 – I suggested above that the first paragraph should go in the introduction, and I think the second paragraph should have its own heading along the lines of "Caveats of ECCO in this region". In addition to this second paragraph, there should also be a list of potential biases in ECCO for this region. The GS/NAC position (especially in the NW Corner

region) is the first one that comes to mind, but also the overflows at Greenland Scotland Ridge is another important one. Another one would be that the shelf circulation is not resolved well in ECCO. ECCO is a great tool but there are caveats that go along with working with it, and the text will read better if you acknowledge those caveats up front rather than let the reader ponder their effects on their own.

Section 4.5 has now been added, where caveats of using ECCO are discussed. While we agree there are some caveats to using ECCO, particularly due to the coarse resolution, we do not believe that any inaccuracies in the model with respect to the Greenland Scotland Ridge overflows or shelf circulation should impact the results of this study. The stratification of the upper ocean in our control volume matches observations well; any potential biases in deep ocean stratification around the Greenland-Scotland ridge do not significantly affect the stratification in our control volume.

16. I found that the text overlooked a relatively well-developed literature that has used ECCOv4 in this region: Piecuch et al. (2017), Foukal and Lozier (2018), Tesdal et al. (2019), and Asbjornsen et al. (2020). Including these papers could strengthen the authors case that ECCO is well-suited for this region.

Thank you for the suggestions, each of these papers have now been cited in the introduction of the paper, and used to discuss the caveats of using ECCO in section 4.5.

Minor edits:

Lines 14-24 – It was a bit strange to repeat the last sentence in the abstract as the second sentence of the introduction. Can you just remove the sentence from the Introduction?

This has now been removed.

Line 77 – what is 1 x 1/3-1? I know ECCO's grid size varies with respect to latitude, but this notation is not clear.

This has now been changed to a "nominal horizontal resolution of 1°".

Line 195 (and throughout) – It's a bit confusing to cite the numbers in the text as °C/month when they're listed as °C/year in the figure (and sometimes in the text). Can you just use one unit throughout?

In the text, all values have now been changed to °C/yr to match the figures.

Fig. 10f – please remove one of the two legends in this panel (but I'll admit two legends are better than no legends!).

Thank you for pointing this out, the second legend has now been removed!

---

## Author Comment (AC2)

**Response to Reviewer 2**

Thank you for your helpful and constructive comments on our paper. The comments are listed below in black, with our response to each in blue. We hope that you are happy with the improved manuscript and find it suitable for publication.

The manuscript presents an in-depth analysis of the onset and evolution of the cold anomaly in the North Atlantic over the 2013-2017 period. Two versions of the ECCOv4 model were used to analyse in detail the different terms of the mixed-layer temperature budget. The new result from this study is the importance of vertical diffusion term in the re-emergence of the SST anomaly in summer, while the advection term is one order of magnitude smaller. This is explained by the strong temperature gradient at the base of the mixed layer, induced by anomalously high surface heat fluxes and subsequent cooling during the previous winters.

The manuscript is very well written, and figures are particularly nice and clear. This is a careful and very interesting study that deserves publication. Considering the temperature temporal variation instead of the usual heat budget is sometimes tricky to understand, but the authors are very careful to accompany the reader through the subtleties of the interpretation. Overall, I have one major remark that needs consideration, and then minor revision comments.

Thank you for the encouraging comments, we hope that you find the responses to your recommendations satisfactory.

My general remark on the method is that the main result bears on the importance of vertical diffusion. However, this is precisely the term parameterized to close the budget, so to my point of view, it needs more than the figure in the appendix to justify this chosen parametrisation. To my knowledge, Kv~2 cm2/s is in the "high range" of values usually used in OGCMs. Could you comment more on this? I explain: imagine that the winter heat fluxes are underestimated in the model. Then the data assimilation will "correct" the insufficient cooling of the mixed layer, and this artefact will necessarily appear in the parameterized term, and I don't understand why the closed mixed layer budgets gives an answer to this problem lines 370-375). I suppose that you have access to the diagnosed Kd in the model: could you make a comparison?

The diffusion at the base of the mixed layer is not comparable with the background mixing in the model; this has now been made clear in the methods section of the manuscript. There are multiple mixing schemes present in the model that represent diffusion, including isopycnal and diapycnal background diffusion, GGL mixed layer turbulence closure, and a convective adjustment for the winter MLD. It is therefore not clear which value(s) the budget diffusivity should be compared against. Instead, we simply choose diffusivity values that are optimal for closing the budget within this specific region. However, the fact that the diffusion term of our budget is still close to that of the mixed layer budget suggests that the values we have chosen do a good job of reproducing the impact of the mixing schemes in the model, and in particular in reproducing the cooling anomalies due to diffusion that are focused on in this paper. The figures showing the closed budgets have now been moved to the main text of the paper (Fig. 5b and 6b) for a clearer comparison between the closed budgets and approximated budgets. We have now discussed the choice of diffusivity further within the methods section.

The closed budget does not answer the problem that the data assimilation could be correcting for insufficient cooling via diffusion. However, the ERA-interim used to force the models have been shown to be consistent with independent observations of surface heat fluxes in the North Atlantic. This has now been further discussed in Section 4.5. However, this is always a caveat when using state estimates to investigate variability.

In the article, the role of horizontal advection is minimized, although it clearly plays two roles in the 2015 cold anomaly origin: onset of the cold anomaly in winter 2013-2014 (25%, line 222) and advection of cold water below the mixed layer during the following summer/autumn (lines 384-386). Although this does not concern directly the processes responsible for the 2015 anomaly, I think it should be more emphasized (in the abstract and conclusion) to contrast with what happened in 2015.

We agree that the impact of advection was poorly expressed in the manuscript. The text and schematics have now been revised to emphasise the role that advection plays in driving the initial cooling of the cold anomaly, as well as the cooling of the sequestered anomaly beneath the mixed layer.

Minor revision:

I think you should merge Fig. 1 and Fig2

Thank you for the suggestion, Fig. 1 and 2 have now been merged.

Fig. 5 and section 3.3: I would put the sum of terms in solid line (as the contributing terms) et and model tendency in dashed line (this is more intuitive).

In all budget figures, the dashed and solid lines have now been switched, so that the budget tendency is shown by the solid line.

Could you add this figure 5 but in the northern and southern box in the appendix (to emphasize the difference in the entrainment term)?

The climatology of the mixed layer heat budget averaged over the northern and southern boxes have now been added to the Appendix (Fig. A3).

lines 189-190: How do you explain that the difference between your budget and the model between May and June? Which assumption is the most problematic to your point of view? Is the possible warming just below the calculated MLD responsible for it?

It's difficult to pinpoint the exact cause of any particular error. The surface flux term is correct for the model as it is calculated in the same way as for the closed budget, however there are sources of error in each of the remaining terms. During summer, it is unlikely that

entrainment is causing the error as the deepening of the mixed layer is still small during this period. It is therefore likely to be the advection and/or diffusion term causing the error, and cooling via diffusion is lower than in the closed budget. It is possible that temperature changes below the mixed layer could be causing some of the error, particularly if the choice of definition for the $\Delta T$ term is not optimal, leading to the underestimate in cooling of the mixed layer.

The most problematic terms of the mixed layer budget equation are the diffusion and entrainment terms as they require the most assumptions/parameterisations, including the parameterisation of diffusivity and the definition for entrainment velocity and $\Delta T$. Despite the error, the similarities between the terms of the mixed layer budget and the equivalent terms within the closed budget, as well as how close the seasonal cycle of the budget is to closure, give us confidence in the method used to approximate the mixed layer budget. The potential reasons for error have now been added to the text (lines 208-216).

Section 3.4: you write that horizontal advection is responsible for about 25% of the cooling in December 2013. However, you disregard this result later on. The advection is more important in the southern box in your analysis (your Fig. 9), so it could confirm the schematics of Holliday et al. (2020), their figure 10. In conclusion, I disagree with the statement lines 383-385: "advection still played only a small role in the initial cooling in comparison to surface forcing". 25-30% is not small.

We agree that the role of advection was wrongly understated. This has now been changed throughout the text to emphasise the role advection plays in the initial forcing of the cold anomaly. Thank you for pointing out the connection between the role of advection and the findings of Holliday et al. (2020); this has now been included in the discussion.

Figure 8: very nice: it concerns mainly 2014, doesn't it? To better understand, please add a figure in the appendix with the absolute values of Fig. 6b so that there is no confusion on the actual sign of each term. Consider adding a small arrow for advection in the first box, and also below the mixed layer in spring and summer to explain why the temperature keeps decreasing (although this is not proven in this ms).

Thank you for the suggestions. The schematic has been edited to include the contribution of advection. The mixed layer budget without the seasonal cycle removed has been added to the Appendix (Fig. A1).

Discussion: Some points are difficult to follow because most figures are anomalies (except diffusion term fig.7c), so we don't know the actual sign of each term. Otherwise, see my main comments above.

Figures showing the absolute values of each budget term, before the seasonal cycle has been removed, have been added to the Appendix (Fig. A1, A4), as well as figures showing the components of the budgets without the seasonal cycle removed (Fig. A2, A5). The sign of each of the terms has also been referenced in the text when applicable, hopefully making the explanations clearer.

---

## Author Response (AR2)

Dear Dr Sanders,

Thanks for taking into account the reviewers remarks very carefully. Your manuscript is (almost) ready for publication. I have however a question. The 11 figures A7-A17 in the appendix are only very briefly mentioned in the text:

"The analysis in ECCOv4-r4 set out in this paper was also repeated in ECCO Version 4 Release 3 (ECCOv4-r3, covering 1992-2015, see Forget et al., 2015a), with the same conclusions reached (Fig. A7-A17)."

Why are 11 figures needed to support this sentence? If these 11 figures are significant, you should explain why by adding text in the appendix. If each of them is not significant, they should be removed.

Please decide what you prefer to do for the final version of your manuscript, regarding these figures.

Best regards,

Anne Marie Treguier

Thank you for the comment. Four of these figures have now been removed from the appendix, with the remaining figures added to Appendix B: Analysis in ECCOv4-r3.  Fig. B1-B7 are a repeat of the figures in the main text but computed using ECCOv4-r3 rather than ECCOv4-r4, in order to give further confidence in our conclusions of the drivers of the 2015 cold anomaly. Fig. B1-B2 show that the 2015 cold anomaly is well-represented in the earlier release of the model. Fig. B3-B7 show that the same processes that are shown to drive the cold anomaly in ECCOv4-r4 are also responsible for driving the anomaly in ECCOv4-r3. Text has now been added to Appendix B explaining the reasons for including these figures.